# Atypical heat shock transcription factor HSF5 is critical for male meiotic prophase under non-stress conditions

Saori Yoshimura[1,2,7], Ryuki Shimada [1,7], Koji Kikuchi [1], Soichiro Kawagoe [3], Hironori Abe[1], Sakie Iisaka[1], Sayoko Fujimura[4], Kei-ichiro Yasunaga[4], Shingo Usuki[4], Naoki Tani[4], Takashi Ohba[2], Eiji Kondoh[2], Tomohide Saio [3], Kimi Araki [5,6] & Kei-ichiro Ishiguro [1] ✉

Meiotic prophase progression is differently regulated in males and females. In males, pachytene transition during meiotic prophase is accompanied by robust alteration in gene expression. However, how gene expression is regulated differently to ensure meiotic prophase completion in males remains elusive. Herein, we identify HSF5 as a male germ cell-specific heat shock transcription factor (HSF) for meiotic prophase progression. Genetic analyzes and single-cell RNA-sequencing demonstrate that HSF5 is essential for progression beyond the pachytene stage under non-stress conditions rather than heat stress. Chromatin binding analysis in vivo and DNA-binding assays in vitro suggest that HSF5 binds to promoters in a subset of genes associated with chromatin organization. HSF5 recognizes a DNA motif different from typical heat shock elements recognized by other canonical HSFs. This study suggests that HSF5 is an atypical HSF that is required for the gene expression program for pachytene transition during meiotic prophase in males.

Meiosis occurs prior to the formation of sperm and oocytes. Meiotic entry is followed by the meiotic prophase, in which meiosis-specific chromosomal events such as chromosome axis formation, homolog synapsis, and meiotic recombination occur sequentially[1–4]. The meiotic prophase is regulated by sexually dimorphic mechanisms, such that the gene expression program is altered for the subsequent developmental program of spermatid morphogenesis or oocyte maturation. In the male meiotic prophase, developmental progression beyond the pachytene stage is a critical event in which multiple gene regulatory programs inactivate gene expression on sex chromosomes, suppress transposable elements, and progress toward the post-meiotic stage[5–8].

Accordingly, a subset of male germ cell-specific transcription factors is required for progression beyond the pachytene stage[5,9–15]. However, it remains unclear how pachytene progression is ensured in the male meiotic prophase and which transcription factors are responsible for this process.

Previously, we identified MEIOSIN, which plays an essential role in meiotic initiation in males and females[16]. MEIOSIN, together with STRA8[17,18], activates numerous meiotic genes. Among the target genes of MEIOSIN whose functions are unknown, we identified new germ cell-specific factors involved in meiosis[11,19,20]. The heat shock transcription factor family 5 (*Hsf5*) gene was also identified as one of the MEIOSIN/STRA8 target genes.

[1]Department of Chromosome Biology, Institute of Molecular Embryology and Genetics (IMEG), Kumamoto University, Honjo 2-2-1, Chuo-ku, Kumamoto 860-0811, Japan. [2]Department of Obstetrics and Gynecology, Faculty of Life Sciences, Kumamoto University, Kumamoto 860-8556, Japan. [3]Division of Molecular Life Science, Institute of Advanced Medical Sciences, Tokushima University, Tokushima 770-8503, Japan. [4]Liaison Laboratory Research Promotion Center, IMEG, Kumamoto University, Kumamoto 860-0811, Japan. [5]Institute of Resource Development and Analysis, Kumamoto University, Kumamoto 860-0811, Japan. [6]Center for Metabolic Regulation of Healthy Aging, Kumamoto University, Kumamoto 860-8556, Japan. [7]These authors contributed equally: Saori Yoshimura, Ryuki Shimada. ✉e-mail: ishiguro@kumamoto-u.ac.jp

The HSF family comprises several paralogs; of these, HSF1, HSF2, HSF4, HSF5, and HSFY are conserved in humans and mice, with an additional HSFX for humans and HSF3 for mice[21]. The mammalian HSF family drives gene regulation events that activate or repress transcription during stress responses and under non-stress conditions[21–23]. Heat stress induces heat shock response (HSR), which is mediated by heat shock proteins (HSPs). The best-known HSF paralog associated with the stress response is HSF1, which is present in the cytoplasm as an inactive monomer when bound to HSPs. Upon sensing stress, HSF1 forms a homotrimer that translocates into the nucleus, binds specifically to the heat shock element (HSE) in the genome, and activates HSR gene transcription. Sarge et al.,[24–26]. Under non-stress conditions, the HSF family is known to regulate developmental processes of spermatogenesis[22]. *Hsf1* knockout (KO) males produce fewer sperms compared to wild-type (WT) mice but are still fertile[27]. *Hsf2* KO males exhibit reduced spermatogenesis but are still fertile[28,29]. *Hsf1* and *Hsf2* double knockout causes male infertility in mice[30]. These suggested that HSF1 and HSF2 play synergistic roles in spermatogenesis under non-stress conditions.

Previous genetic studies have suggested that HSF5 orthologs are involved in spermatogenesis in various species. In zebrafish, *Hsf5* mutant males were infertile with reduced sperm count, increased sperm head size, and abnormal tail architecture, whereas females remained fertile[31]. In human testes, patients with azoospermia and low modified Johnson scores were associated with low expression of HSF5[32]. Additionally, it has been demonstrated that disruption of *Hsf5* led to apoptosis during spermatogenesis in mice, resulting in the failure of meiotic sex chromosome inactivation (MSCI) and consequent infertility[33]. However, it remains unclear which processes of meiosis involve HSF5 and whether HSF5 plays overlapping and/or distinct roles compared to other HSFs in the progression of meiotic prophase in the testis. Furthermore, despite the presence of a DNA-binding domain, it is yet to be determined whether HSF5 acts as a transcription factor under stress or non-stress conditions like other HSFs.

Here, we show that mouse HSF5 plays an essential role in the meiotic prophase progression in male germ cells under non-stress conditions. Our genetic analysis of *Hsf5* KO mice demonstrated that HSF5 was required for progression beyond the pachytene stage during spermatogenesis. Furthermore, chromatin immunoprecipitation sequencing (ChIP-seq) of HSF5 in vivo combined with DNA-binding analysis in vitro demonstrated that HSF5 binds to the promoters of a subset of genes whose biological functions are associated with chromatin organization through a DNA motif that is different from the typical HSE. The present study suggests that HSF5 acts as an atypical HSF under non-stress conditions that execute the gene expression program for pachytene transition during male meiotic prophase.

## Results

### Identification of *Hsf5* in mice

The *Hsf5* gene was identified as one of the MEIOSIN/STRA8-target genes[16] (Fig. 1a). However, apart from HSF5 being potentially required in the establishment of male MSCI, its biological function in mice has yet to be fully elucidated[33].

Six paralogous genes encoding HSFs were identified in the mouse genome (Fig. 1b). The HSF family possesses a DNA-binding domain and acts as a transcription factor in the heat stress response (HSR) and under non-stress conditions[21,22]. HSF5 possesses a winged-helix-turn-helix (WTHH)-like DNA-binding domain (Fig. 1b). While HSF1, HSF2, and HSF4 possess two heptad repeats, HR-A and HR-B, which are predicted to form inter-molecular leucine zippers for homotrimer oligomerization[34], HSF5 lacks these heptad repeats (Fig. 1b).

Size exclusion chromatography-multi angle light scattering (SEC-MALS) analysis demonstrated that the purified His-MBP-fused full-length HSF5 protein eluted at peaks corresponding to approximate molar masses of 160 kDa and 340 kDa at 25 °C, as well as at the peak with megadalton-scale size after treatment at 42 °C (Fig. 1c). This observation indicates that while HSF5 exists as a mixture of monomers and dimers/trimers at 25 °C, it forms high-order oligomers at higher temperatures. This thermal response of HSF5 resembles HSF1 which forms higher-order oligomers upon heat shock[35]. In contrast, the HSF5 fragment (aa 1-209) containing DNA binding domain (HSF5-DBD) was in a monomeric state at both 25 °C and 42 °C, indicating that oligomerization of HSF5 is mediated by another region than DBD. Thus, HSF5 exhibits a structural feature to undergo oligomerization in vitro.

### HSF5 is expressed in the spermatogenic lineage in the testis

We examined the steady-state mRNA levels of *Hsf5* in different mouse tissues using RT-PCR. *Hsf5* mRNA was specifically detected in the juvenile and adult mouse testes but not in other adult organs we examined (Fig. 1d), which is in stark contrast to the ubiquitous steady-state mRNA abundance of *Hsf1*, *Hsf2*, and *Hsf4* (Fig. 1d, Supplementary Fig. 1a). Similarly, *Hsfy2* mRNA was detected in the juvenile and adult testes. Although the steady-state mRNA abundance of mouse *Hsf3* in any organ is unknown in the available database (Supplementary Fig. 1a), our RT-PCR showed *that Hsf3* is detected in the testes and ovaries. The spermatogenic mRNA levels of *Hsf5* and other *Hsf* members were assessed by reanalyzing scRNA-seq data from adult mouse testes[36] (Supplementary Fig. 1b). The Uniform Manifold Approximation and Projection (UMAP) of scRNA-seq data showed that *Hsf5* mRNA levels were increased in meiotic prophase, and different from those of other *Hsf* members in mouse spermatogenic cells (Supplementary Fig. 1b). In contrast to spermatogenic expression, *Hsf5* expression was hardly detected in embryonic day 12.5 (E12.5) – E18.5 fetal ovaries by RT-qPCR (Fig. 1e), suggesting *Hsf5* was expressed at a low level in fetal ovaries. These observations suggest that HSF family paralogs function at different stages in the testis and that HSF5 plays a meiosis-specific role during spermatogenesis.

To determine the stage when HSF5 protein is specifically expressed, seminiferous tubules of the WT mouse testes (8 weeks old) were immunostained with specific antibodies against HSF5 along with SYCP3 (a component of meiotic axial elements) and γH2AX (a marker of DSBs and XY body) or STRA8 (a marker of preleptonema) (Fig. 2a). The HSF5 signal began to appear in the mid-pachytene spermatocyte nuclei of the stage VI seminiferous tubules, and was observed in the spermatocyte nuclei of the stage VII-XII, and in the round spermatids of stage I-VI (Fig. 2a). However, HSF5 was not observed in spermatogonia or spermatocytes before mid-pachytene or in elongated spermatids (Fig. 2a). The same immunostaining patterns of the HSF5 signal were confirmed by other HSF5 antibodies (HSF5-N1 and HSF5-N2) (Supplementary Fig. 2a).

To further validate the expression pattern of HSF5 in the testis, we generated *Hsf5-3xFLAG-HA* knock-in mice (Fig. 2b) and verified the expression of the fusion protein in the testis extracts by immunoblotting (Fig. 2c). The same pattern was further verified by immunostaining using the HA antibody in the seminiferous tubules of *Hsf5-3xFLAG-HA* knock-in mice (Fig. 2d). Close inspection verified that HSF5 was detected in the nuclei of spermatocytes and round spermatids (Supplementary Fig. 2b, c). We noticed that an intense HSF5 signal appeared in pachytene spermatocytes, particularly at stages VII–VIII, which coincided with the γH2AX signal (Fig. 2a, d, Supplementary Fig. 2a). This observation was further validated using pachytene spermatocytes squashed from the stage VII–VIII seminiferous tubules excised from *Stra8-3xFLAG-HA-p2A-GFP* knock-in mice[16] (Supplementary Fig. 2d). Although we reproducibly observed the intense HSF5 signal associated with the XY chromosomes in pachytene spermatocytes at stages VII–VIII on seminiferous tubule sections, close inspection revealed that ~11.8% (*n* = 204) of pachytene spermatocytes showed such HSF5 signals associated with the XY chromosomes (Supplementary Fig. 2d). Furthermore, confocal microscopy analysis

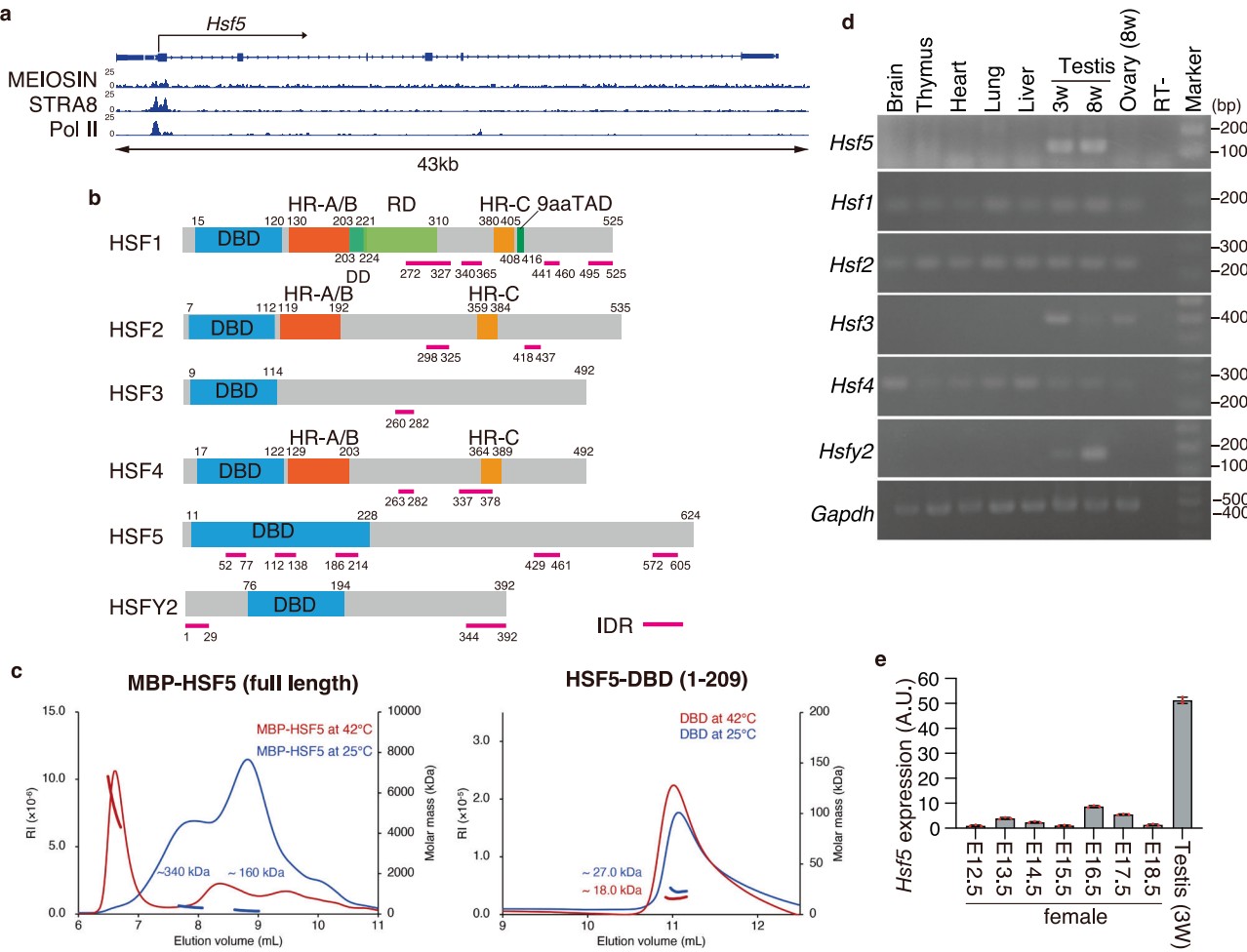

**Fig. 1 | HSF5 is expressed during meiotic prophase in the mouse testis.**
**a** Genomic view of MEIOSIN and STRA8 binding peaks over the *Hsf5* locus. Genomic coordinates derived from NCBI. To specify testis specific transcription, RNA polymerase II ChIP-seq in the testis is shown[5]. **b** Schematic diagram of domain structure in six members of mouse heat shock factor (HSF) protein family. Domain name and the number of the amino acid residues are assigned according to Uniprot. DBD DNA-binding domain, HR heptad repeat, IDR Intrinsically disordered region, DD D domain, RD Regulatory domain, 9aaTAD; Transactivation motif. **c** SEC-MALS profiles of MBP-full length HSF5 (left) and HSF5-DNA binding domain (a.a. 1-209) (right) at room temperature (blue) and after heat treatment at 42 °C for 30 min (red). Thin and bold lines show the refractive index (RI) profile and mass plot, respectively. **d** The tissue-specific expression pattern of *Hsf5* was examined by RT-PCR. Testis RNA was obtained from 3 weeks old (3w) and 8 weeks old (8w) male mice. Ovary RNA was obtained from adult 8 weeks old (8w) female mice. RT-indicates control PCR without reverse transcription. The data was acquired from two separate experiments. **e** The expression patterns of *Hsf5* in the embryonic ovary (E12.5-E18.5, *n* = 1 for each) and testis (3w, *n* = 1) were examined by RT-qPCR. Average values normalized to E12.5 ovary are shown with SD from technical triplicates.

using whole-mount immunostaining of the stage VII–VIII seminiferous tubules revealed that HSF5 signals were predominantly skewed on the chromatin loops rather than on the axes (Supplementary Fig. 2e). One explanation is that HSF5 may form a labile aggregate around the XY body rather than tightly localizing on the XY chromosomes in pachytene spermatocytes at stages VII–VIII. Testis-specific histone H1t is a marker of spermatocytes later than mid-pachytene and round spermatids[37,38]. Immunostaining of seminiferous tubules at postnatal day 16 (P16) by H1t along with HSF5 indicated that the HSF5 signal started to appear following the expression of H1t (Fig. 2e). These observations suggest that HSF5 is involved in developmental regulation at the mid-to-late-pachytene stage of meiotic prophase onward in males (Fig. 2f).

**Spermatogenesis was impaired in *Hsf5* knockout males**
To address the role of HSF5 in mice, we deleted all coding exons (Exon1-Exon6) of the *Hsf5* loci in C57BL/6 fertilized eggs using the CRISPR/Cas9 system (Fig. 3a). Immunoblotting of the extract from *Hsf5* knockout (KO) testes showed that the HSF5 protein was absent

(Fig. 3b), which was further confirmed by the diminished immunolocalization of HSF5 in the seminiferous tubules of *Hsf5* KO mice (Fig. 3c), indicating that the targeted *Hsf5* allele was knocked out.

Although *Hsf5* KO male mice did not show overt phenotypes in somatic tissues, examination of the reproductive organs revealed smaller testes in *Hsf5* KO mice compared to those in WT and *Hsf5* +/- mice during the juvenile (4 weeks old) and adult (8 weeks old) periods (Fig. 3d). Histological analysis revealed that post-meiotic spermatids and spermatozoa were absent in 4 weeks and 8 weeks old *Hsf5* KO mice, in contrast to their WT and *Hsf5* +/- siblings (Fig. 3e). Accordingly, sperm were absent from the adult *Hsf5* KO caudal epididymis at 8 weeks (Fig. 3f). Consistently, seminiferous tubules containing PNA lectin (a marker of spermatids)-positive cells were absent in *Hsf5* KO mice (Fig. 3g). Thus, the later stages of spermatogenesis were abolished in *Hsf5* KO seminiferous tubules, resulting in male infertility (Fig. 3h). In contrast to males, *Hsf5* KO females exhibited normal fertility with no apparent defects in the adult ovaries (Supplementary Fig. 3a–c). These results reveal that HSF5 is essential for spermatogenesis but not oogenesis.

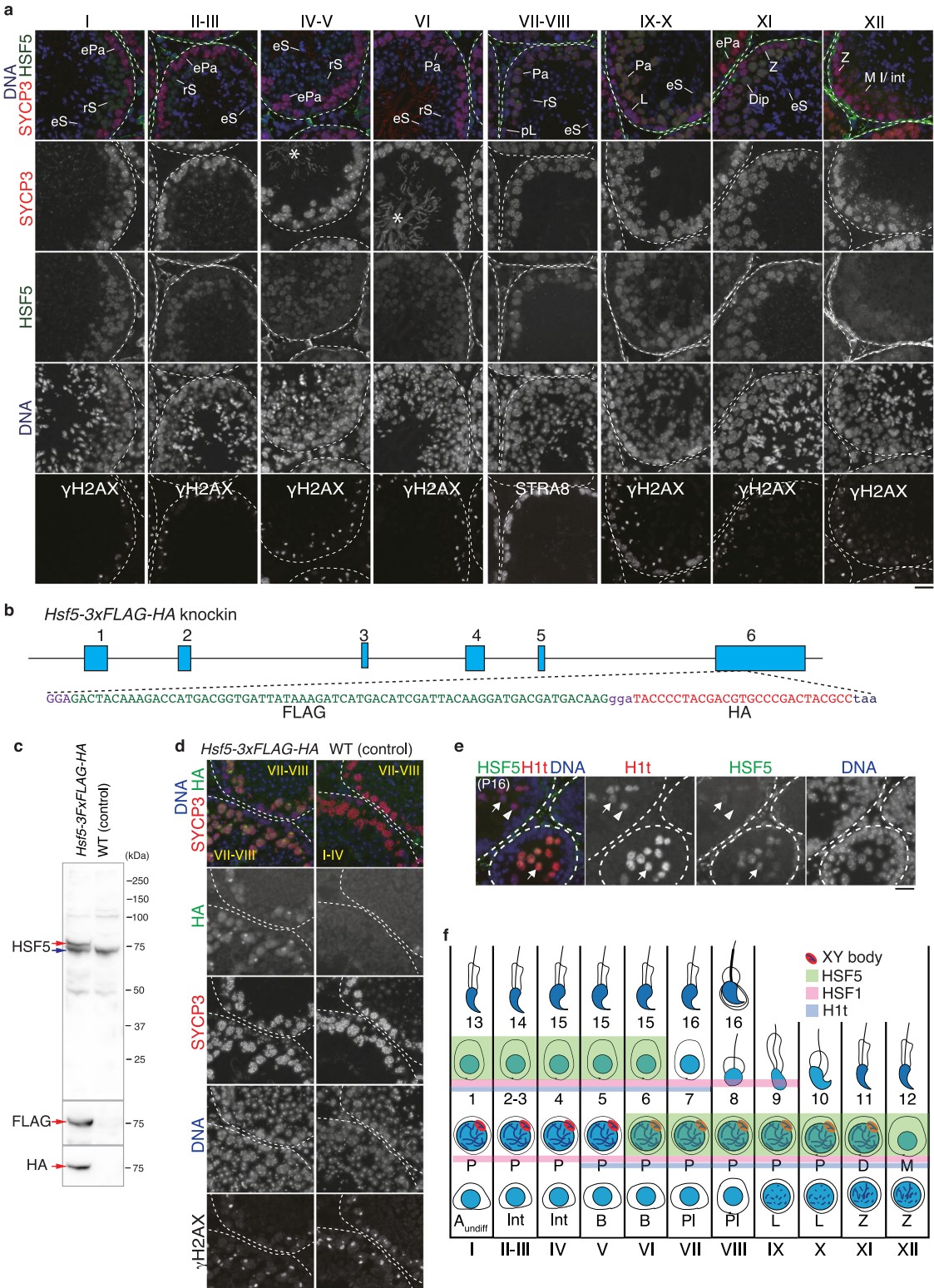

## HSF5 is required for progression through pachytene in male meiotic prophase

To identify the stage at which the primary defect appeared in the *Hsf5* KO, we analyzed the progression of spermatogenesis by immunostaining seminiferous tubules (4 weeks old) with antibodies against SYCP3 and H1t (a marker of spermatocytes later than mid-pachytene and round spermatids). While all the seminiferous tubules in WT contained H1t-positive spermatocytes and/or round spermatids at

4 weeks old, H1t-positive spermatocytes, but not round spermatids, were observed in around 55% (*n* = 3) of total seminiferous tubules in the same aged *Hsf5* KO (Fig. 4a). In *Hsf5* KO seminiferous tubules, fewer H1t-positive spermatocytes, if any, were observed and were accompanied by an aberrant staining pattern of SYCP3. These observations suggest that *Hsf5* KO spermatocytes reached at least the mid-pachytene stage, but the progression of meiotic prophase beyond pachytene was compromised in the absence of HSF5.

**Fig. 2 | HSF5 expression pattern in the mouse seminiferous tubules.**
**a** Seminiferous tubule sections in WT testis (8 weeks old) were immunostained as indicated. pL preleptotene, L Leptotene, Z Zygotene, ePa early Pachytene, P Pachytene, M I Metaphase I, int Interkinesis, rS round Spermatid, eS elongated Spermatid. Boundaries of the seminiferous tubules are indicated by white dashed lines. * indicates a non-specific cross-reactivity of the gunia pig anti-SYCP3 antibody to sperm tail. HSF5 was immunostained by HSF5-C antibody. The same immunostaining pattern of HSF5 was confirmed by other HSF5-N1 and HSF5-N2 antibodies, as shown in Fig. S2a. Roman numbers indicate the seminiferous tubule stages. Biologically independent mice ($n = 3$) were examined in three separate experiments. Scale bar: 25 μm. **b** The schematic of the *Hsf5-3xFLAG-HA* knockin allele. **c** Testis extracts from *Hsf5-3xFLAG-HA* knockin and negative control WT mouse testis (5 weeks old) were immunoblotted as indicated. Red arrow indicates HSF5-3xFLAG-HA protein derived from the knockin allele. Blue arrow indicates HSF5 protein derived from WT allele. **d** Seminiferous tubule sections in *Hsf5-3xFLAG-HA* knockin ($n = 1$) and negative control WT mouse testis ($n = 1$) at 5 weeks old were immunostained as indicated. Scale bar: 25 μm. **e** Seminiferous tubule sections in WT testis (P16) were immunostained as indicated. Arrow and arrowhead indicate HSF5-positive/H1t-positive and HSF5-negative/H1t-positive pachytene spermatocytes, respectively. Scale bar: 25 μm. A single experiment was performed. **f** The schematic of expression of HSF5 (green), H1t (blue), HSF1(red)[46], in the stages of the seminiferous tubules.

Immunostaining analysis with antibodies against SYCP3 and SYCP1 (markers of homolog synapsis) and γH2AX demonstrated that spermatocytes underwent homologous chromosome synapsis in juvenile *Hsf5* KO males (P21), as in age-matched controls (Fig. 4b). However, spermatocytes after pachytene and post-meiotic spermatids were not observed in *Hsf5* KO mice at P21 (Fig. 4b). Accordingly, more leptotene/zygotene and reciprocally fewer pachytene populations were observed in *Hsf5* KO spermatocytes than in age-matched controls, suggesting that *Hsf5* KO spermatocytes were arrested at pachytene.

Normally, the first wave of γH2AX is mediated by ATM after double-strand break (DSB) formation in leptotene[39] and disappears during DSB repair. The second wave of γH2A in the zygotene stage is mediated by ATR, which targets unsynapsed or unrepaired chromosomes[40]. At the leptotene and zygotene stages, γH2AX signals appeared in *Hsf5* KO spermatocytes in the same manner as in WT (Fig. 4b), indicating that DSB formation occurred normally in *Hsf5* KO spermatocytes. We observed that ~75.8% of *Hsf5* KO pachytene spermatocytes exhibited a typical γH2AX signal on the XY body (Fig. 4b). Importantly, XY body formation, defined by the γH2AX domain at the periphery of the nucleus, coincided with the exclusion of RNA polymerase II (pol II) from the XY body in all examined spermatocytes (*Hsf5* +/-, $n = 60$, and *Hsf5* KO, $n = 55$ spermatocytes, respectively) (Fig. 4c). Thus, *Hsf5* KO pachytene spermatocytes continued to form the XY body, indicating the presence of MSCI characterized by chromosome-wide silencing of the XY[41].

However, atypical γH2AX staining patterns in *Hsf5* KO pachytene spermatocytes were observed. Specifically, γH2AX signals largely persisted throughout the nuclei, including on fully synapsed autosomes, in ~24.2% of *Hsf5* KO pachytene spermatocytes ($n = 62$), whereas they disappeared in the control pachytene spermatocytes except for retaining on the XY body. Furthermore, BRCA1, a marker of DNA damage response[42,43], appeared along autosomes in ~17.8% of *Hsf5* KO pachytene spermatocytes ($n = 62$) (Fig. 4d). These observations suggested that DSBs were not repaired and/or were newly generated in *Hsf5* KO spermatocytes. Consistently, the number of DMC1 foci (a marker of ssDNA at the DBS site) was significantly increased in *Hsf5* KO pachytene spermatocytes, suggesting that DSBs were yet to be fully repaired in some, if not all, *Hsf5* KO pachytene spermatocytes (Fig. 4e). Accordingly, the number of MLH1 foci (a marker of crossover recombination) was reduced in *Hsf5* KO pachytene spermatocytes compared to that in the control (Fig. 4f), suggesting that crossover recombination was incomplete in *Hsf5* KO pachytene spermatocytes. These results suggest that DBS repair and crossover formation are defective in *Hsf5* KO pachytene spermatocytes despite fully synapsed homologs.

Notably, a higher number of TUNEL-positive seminiferous tubules (~19.3% of total tubules) was observed in *Hsf5* KO testes at 4 weeks (Fig. 4g). Since TUNEL-positive cells were observed in stage VI seminiferous tubules that contained mid-pachytene spermatocytes in *Hsf5* KO mice (Fig. 4g), ongoing germ cell degeneration presumably occurred at mid-pachytene in *Hsf5* KO spermatocytes, at the time when HSF5 first appeared during spermatogenesis (Fig. 2a, f).

These observations suggest that *Hsf5* KO spermatocytes failed to progress through the pachytene stage and were consequently eliminated by apoptosis. Therefore, HSF5 is required for progression through the pachytene stage of the meiotic prophase.

## HSF5 is dispensable for HSR in testes

To assess whether HSF5 involves in the HSR, gene expression changes in the *Hsf5* +/- and *Hsf5* KO testes were compared at 33 and 37 °C. Whole testes from the *Hsf5* +/- and *Hsf5* KO mice were incubated at 33 or 37 °C for 3 h, and their transcriptomes were examined by RNA-seq. Principal component analysis (PCA) indicated that *Hsf5* +/- and *Hsf5* KO testes exhibited the same trend of differential gene expression at 37 °C versus 33 °C along PC3 (Fig. 5a). Moreover, 39 and 417 genes were identified as differentially expressed genes (DEGs) in the *Hsf5* +/- and *Hsf5* KO, respectively (Fig. 5b, Supplementary Data 2). Notably, 33 DEGs were commonly identified in the *Hsf5* +/- and *Hsf5* KO, including the HSR genes *Hspa1a* and *Hspa1b* (Fig. 5c, d, Supplementary Data 2). These data indicate that HSF5 is dispensable for heat stress response in the testis.

## *Hsf5* KO spermatocytes showed alteration of gene expression at meiotic prophase

HSF1 and HSF2 are implicated in spermatogenesis under non-stress conditions[27–30]. Given that *Hsf5* KO spermatocytes failed to progress beyond the pachytene stage under non-stress conditions in our cytological analyzes (Fig. 4), we conducted transcriptome analysis to determine whether *Hsf5* KO spermatocytes had altered gene expression profiles under non-stress conditions. For this purpose, we isolated spermatocytes that were in the progression of meiotic prophase by fluorescent sorting with DyeCycle Violet (DCV) staining from WT and *Hsf5* KO testes (Supplementary Fig. 4a, b)[44]. Since H1t-negative early pachytene is the stage before defects appeared in the mutants (Fig. 4a), we assumed that the cellular composition should be similar in the control WT and *Hsf5* KO until the first wave of meiotic prophase reaches the mid-pachytene stage. This allowed for the comparison of the transcriptomes of the sorted cells in WT and *Hsf5* KO mice with minimized batch effects that could potentially be caused by a bias in the cellular population. Since the number of sorted populations was limited, we conducted SMART RNA-seq on the sorted spermatocytes, which allowed for RNA-seq analysis with small cell numbers (Supplementary Fig. 4).

PCA revealed that the overall transcriptomes of enriched spermatocytes in *Hsf5* KO testes differed from those in WT testes (Supplementary Fig. 4c). Among the DEGs between WT and *Hsf5* KO pachytene spermatocytes, 958 genes were downregulated in *Hsf5* KO mice, whereas 14 genes were upregulated in *Hsf5* KO mice (Supplementary Fig. 4d). Gene enrichment analysis indicated that the genes involved in cilium movement, male gamete generation, and sperm axonemal dynein complex assembly were downregulated in *Hsf5* KO mice (Supplementary Fig. 4e, Supplementary Data 3). In contrast, the genes upregulated in *Hsf5* KO mice (14 genes) were not significantly associated with any Gene enriched terms. Reanalysis of the

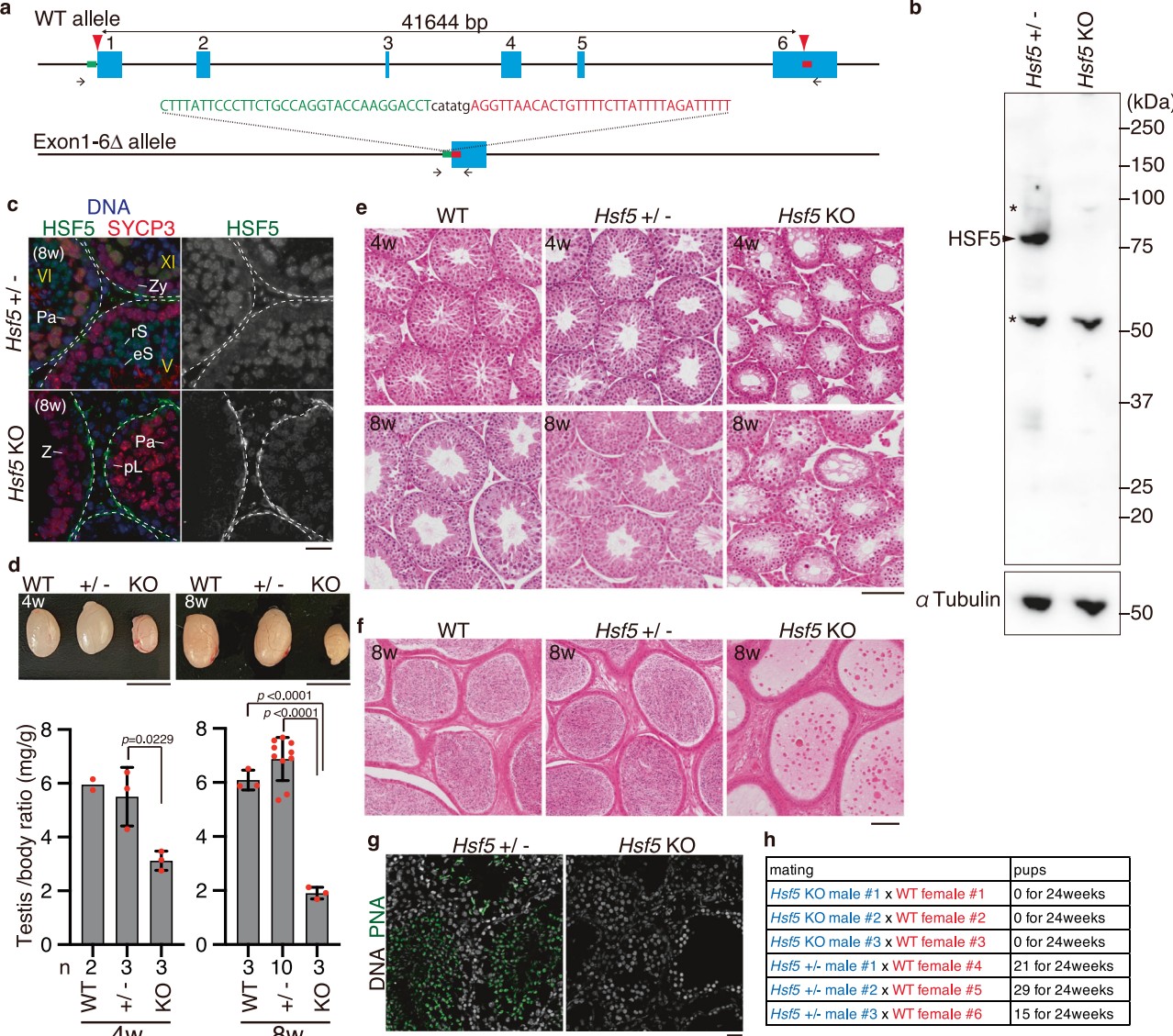

**Fig. 3 | Spermatogenesis was impaired in *Hsf5* knockout male. a** The targeted *Hsf5* allele with deletion of Exon1-6 is shown. 5'- and 3'- homology sequences in the ssODN are shown in green and red, respectively. Arrowheads: synthetic gRNAs. Arrows: PCR primers for genotyping. **b** Immunoblot analysis of testis extracts prepared from mice with the indicated genotypes (P17). The arrow indicates a band of HSF5. * indicates nonspecific bands. Two technically independent experiments from two pairs of WT and *Hsf5* KO siblings were repeated and showed similar results. **c** Seminiferous tubule sections (8 weeks old) were stained for SYCP3, HSF5, and DAPI. pL preleptotene, Pa pachytene spermatocyte, rS round spermatid, eS elongated spermatid. Boundaries of the seminiferous tubules are indicated by white dashed lines. Roman numbers indicate the seminiferous tubule stages. Biologically independent mice (*n* = 3) for each genotype were examined. Scale bars: 25 μm. **d** Testes from WT, *Hsf5* +/- and *Hsf5* KO (upper left: 4 weeks old, upper right: 8 weeks old left). Testis/body-weight ratio (mg/g) of WT, *Hsf5* +/-, and *Hsf5* KO mice (lower left: 4 weeks old, lower right: 8 weeks old) are shown below (Mean with SD). n: the number of animals examined. Statistical significance is shown by *p* value (Two-tailed t-test). Scale bar: 5 mm. **e** Hematoxylin and eosin staining of the sections from WT, *Hsf5* +/- and *Hsf5* KO testes (upper: 4 weeks old, lower: 8 weeks old). Biologically independent mice (*n* = 3) for each genotype were examined. Scale bar: 100 μm. **f** Hematoxylin and eosin staining of the sections from WT, *Hsf5* +/- and *Hsf5* KO epididymis (8 weeks old). Biologically independent mice (*n* = 3) for each genotype were examined. Scale bar: 100 μm. **g** Seminiferous tubule sections (8 weeks old) were stained for PNA lectin and DAPI. Scale bar: 25 μm. A single experiment was performed. **h** Number of pups born by mating *Hsf5* + /- and *Hsf5* KO males with WT females to examine fertility. *Hsf5* +/- males and *Hsf5* KO males were initially mated with WT females (all 4 weeks old at the start point of mating). This cage was observed for 24 weeks from the start of mating.

downregulated genes in *Hsf5* KO (958 genes) using previously published scRNA-seq data from spermatogenic cells[36] suggested that these genes were expressed around the mid-stage of pseudotime and declined after that (Supplementary Fig. 4f). Thus, these results suggest that the gene expression pattern is different between WT and *Hsf5* KO spermatocytes during meiotic prophase, although it is still possible that subtle differences in developmental cellular populations between control WT and *Hsf5* KO spermatocytes may have potentially contributed to bulk transcriptomic differences.

## Failure of developmental progression beyond a substage of mid-pachytene in *Hsf5* KO spermatocytes

Since the pachytene stage is solely defined by the cytologically defined appearance of chromosomal morphology (spermatocytes with fully synapsed homologs) and covers a broad developmental period, it remains unclear whether a specific subtype of pachytene spermatocytes was eliminated during spermatogenesis, or whether a newly emerged subtype caused the primary defect in *Hsf5* KO mice. To identify which subtypes of *Hsf5* KO spermatocytes accompanied

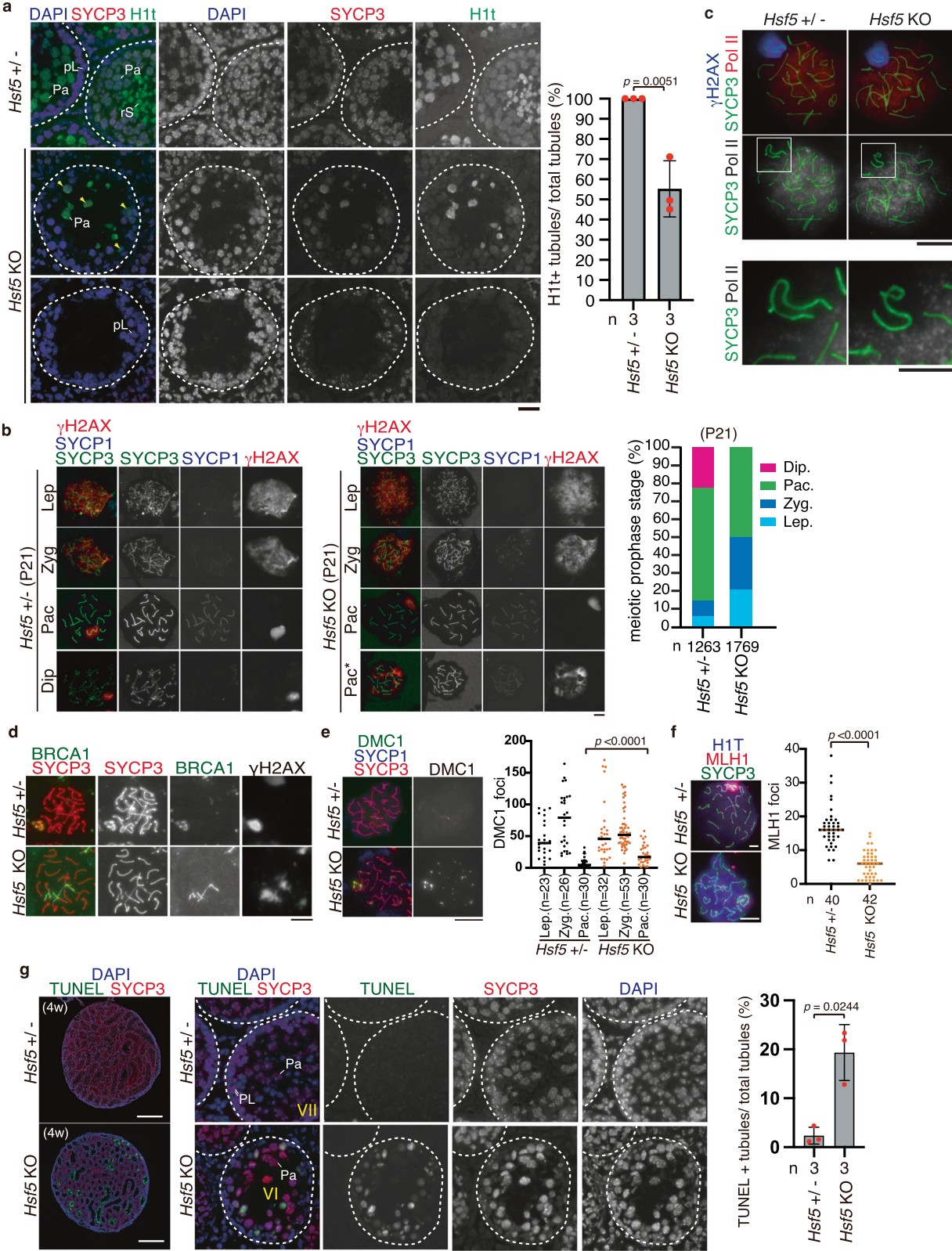

the alteration of gene expression profiles, we further conducted scRNA-seq analyses of whole testicular cells from WT and *Hsf5* KO mice at P16 (Supplementary Fig. 5). Although the cellular composition in the testes changes with developmental progression, we assumed that the first wave of spermatocytes would progress with the same cellular composition in WT and *Hsf5* KO mice until mid-pachytene. We sought a subtype of spermatocytes that would be present in WT but not in *Hsf5*

KO testes, or vice versa, predicted from gene expression patterns at this time point, because it would approximately cover the transition when HSF5 started to exhibit its mandatory function in WT spermatocytes and when the primary defect appeared in *Hsf5* KO spermatocytes. Since HSF5 expression was restricted to germ cells, scRNA-seq datasets derived from germ cell populations (spermatogonia and spermatocytes) were analyzed separately from those of testicular

**Fig. 4 | Hsf5 KO spermatocytes failed to progress through pachytene.**
**a** Seminiferous tubule sections (4 weeks) were stained as indicated. Scale bar: 25 μm. Yellow arrowhead indicates H1t-positive spermatocyte with abnormal SYCP3 staining. pL: preleptotene, Pa: pachytene, rS: round spermatid. Shown on the right is the quantification of the seminiferous tubules that have H1t + /SYCP3+ cells per the seminiferous tubules that have SYCP3+ spermatocyte cells in *Hsf5* +/- and *Hsf5* KO testes (Mean with SD). n: the number of animals examined. Statistical significance is shown (p = 0.0051, unpaired two-tailed t-test). **b**–**f** Chromosome spreads of *Hsf5* +/- and *Hsf5* KO spermatocytes (P21) were immunostained as indicated. **b** Lep: leptotene, Zyg: zygotene, Pac: pachytene, Dip: diplotene. Pac* indicates pachytene spermatocyte with high level of γH2AX signals remained on autosomes. Scale bar: 10 μm. Shown on the right is quantification of stages per total SYCP3+ spermatocytes. n: the number of cells examined. **c** *Hsf5* +/- (n = 60) *Hsf5* KO (n = 55). Scale bar: 10 μm. Enlarged images of the XY body are shown on the bottom. Scale bar: 5μm. **d** ~17.8% of *Hsf5* KO pachytene spermatocytes (n = 62) exhibited

BRCA1 along autosomes with γH2AX signals, whereas none of *Hsf5* +/- pachytene spermatocytes (n = 51) did except for XY chromosome. Scale bar: 10 μm. **e** The number of DMC1 foci is shown in the scatter plot with median (right). Statistical significance is shown (p < 0.0001, two-sided Mann-Whitney U-test). Lep leptotene, Zyg Zygotene, Pac Pachytene. Scale bar: 10 μm. **f** The number of MLH1 foci is shown in the scatter plot with median (right). Statistical significance is shown (Mann-Whitney U-test). n: number of spermatocytes examined. Statistical significance is shown (p < 0.0001, two-sided Mann–Whitney U-test). Scale bar: 5 μm.
**g** Seminiferous tubule sections (4 weeks) were subjected to TUNEL assay. Whole testis sections (left, Scale bar: 500 μm) and closeup view of seminiferous tubule sections (middle, Scale bar: 25 μm) are shown. Shown on the right is the quantification of the seminiferous tubules that have TUNEL+ cells per total tubules in *Hsf5* +/- (n = 3) and *Hsf5* KO (n = 3) testes (Mean with SD). Statistical significance is shown by *p*-value (p = 0.0244, unpaired two-tailed t-test).

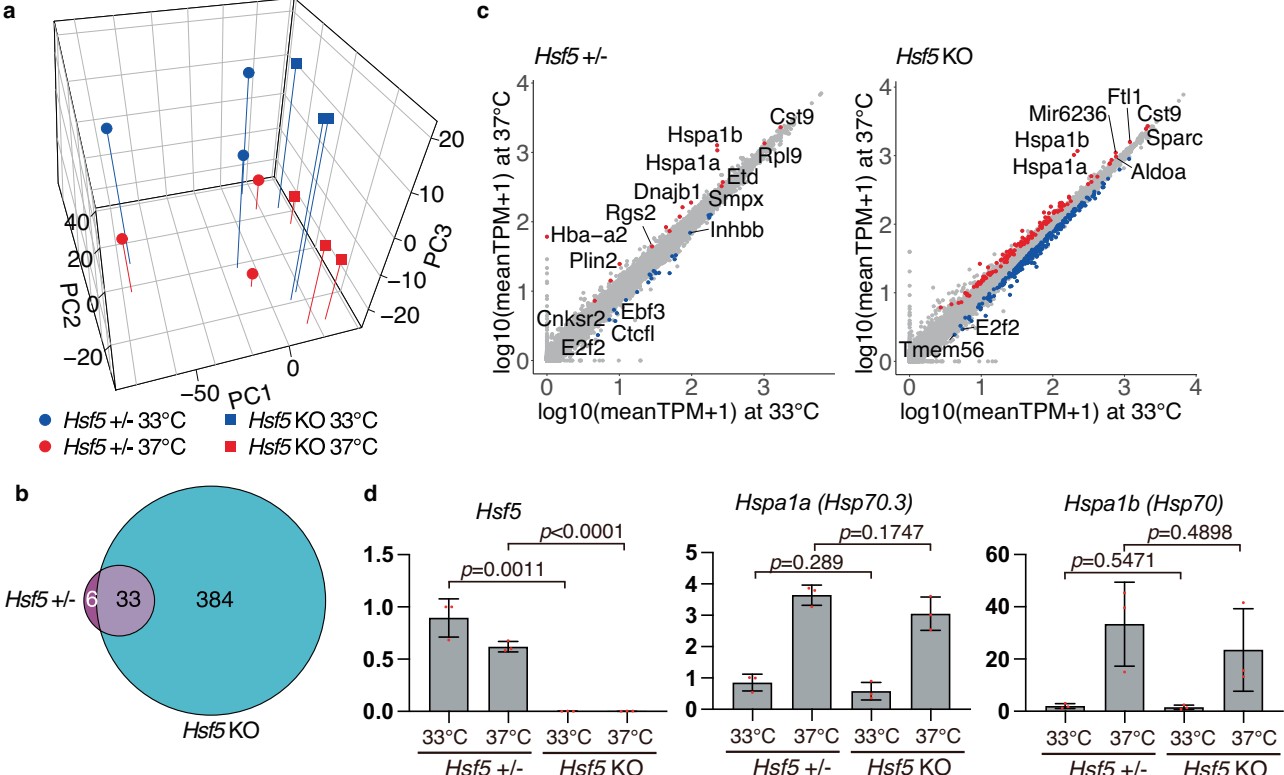

**Fig. 5 | RNA-seq analysis of whole testes at 33 and 37 °C in Hsf5 KO. a** Principal component analysis of the transcriptomes of whole testes from *Hsf5* +/-(n = 3) and *Hsf5* KO (n = 3) at P16, that were incubated at 33 and 37 °C for 3 h. **b** Venn diagram representing the overlap of DEGs at 33 versus 37 °C between *Hsf5* +/- testes and *Hsf5* KO testes. **c** Shown are scatter plot of the transcriptomes of *Hsf5* +/- (n = 3, left) or *Hsf5* KO (n = 3, right) testes treated at 33 °C versus at 37 °C. The red dots indicate

the upregulated genes and the blue dots indicate the downregulated genes at 37 °C. **d** The expression changes of *Hsf5, Hspa1a(Hsp70.3)* and *Hspa1b(Hsp70)* at 33 °C and 37 °C by in the *Hsf5* +/- and KO testes (P16, n = 3) were examined by RT-qPCR. Average values normalized to *Hsf5* +/- testis at 33 °C are shown with SD from technical triplicates. Statistical significance is shown by *p*-value (unpaired two-tailed t-test).

somatic cells (Sertoli cells, Leydig cells, peritubular myoids, endothelial cells, and hemocytes) (Fig. 6a, Supplementary Fig. 5a, b).

The UMAP of the scRNA-seq dataset indicated that the gene expression patterns of single cells isolated from WT and *Hsf5* KO spermatogenic germ cell populations were separated into 12 clusters (Fig. 6b, c, Supplementary Fig. 5c, d, and Supplementary Data 4). The expression patterns of key marker genes were used to estimate the developmental direction of spermatogenesis at the P16 on the UMAP (Fig. 6d). Cluster 11 represented the SSC population, as suggested by the high expression *Gfra1* and *Zbtb16*. Cluster 8 represented an undifferentiated spermatogonial population, as suggested by the high expression *Zbtb16*. Cluster 4 represented the population of spermatogonia A. Cluster 7 represented the population of differentiating

spermatogonia, as suggested by the upregulation of *Kit*. Clusters 5 and 6 represented the population of spermatogonia B. Cluster 1 represents the population at meiotic initiation, as suggested by the upregulation of *Meiosin*. Clusters 0 and 9 represented the early meiotic prophase population, as suggested by the upregulation of *Spo11*. Clusters 3, 2, and 10 represented the mid-meiotic prophase populations, as suggested by the upregulation of *Tesmin*. Based on these, we assumed that spermatogenesis progressed along the trajectory from Cluster 11 to Cluster 10 in the UMAP.

The gene expression profiles of WT and *Hsf5* KO germ cell populations were well overlapped along the trajectory from Clusters 11 to 3. However, we noticed subpopulations (Clusters 2 and 10) that were present in WT but largely missing in *Hsf5* KO testes (Fig. 6b, c),

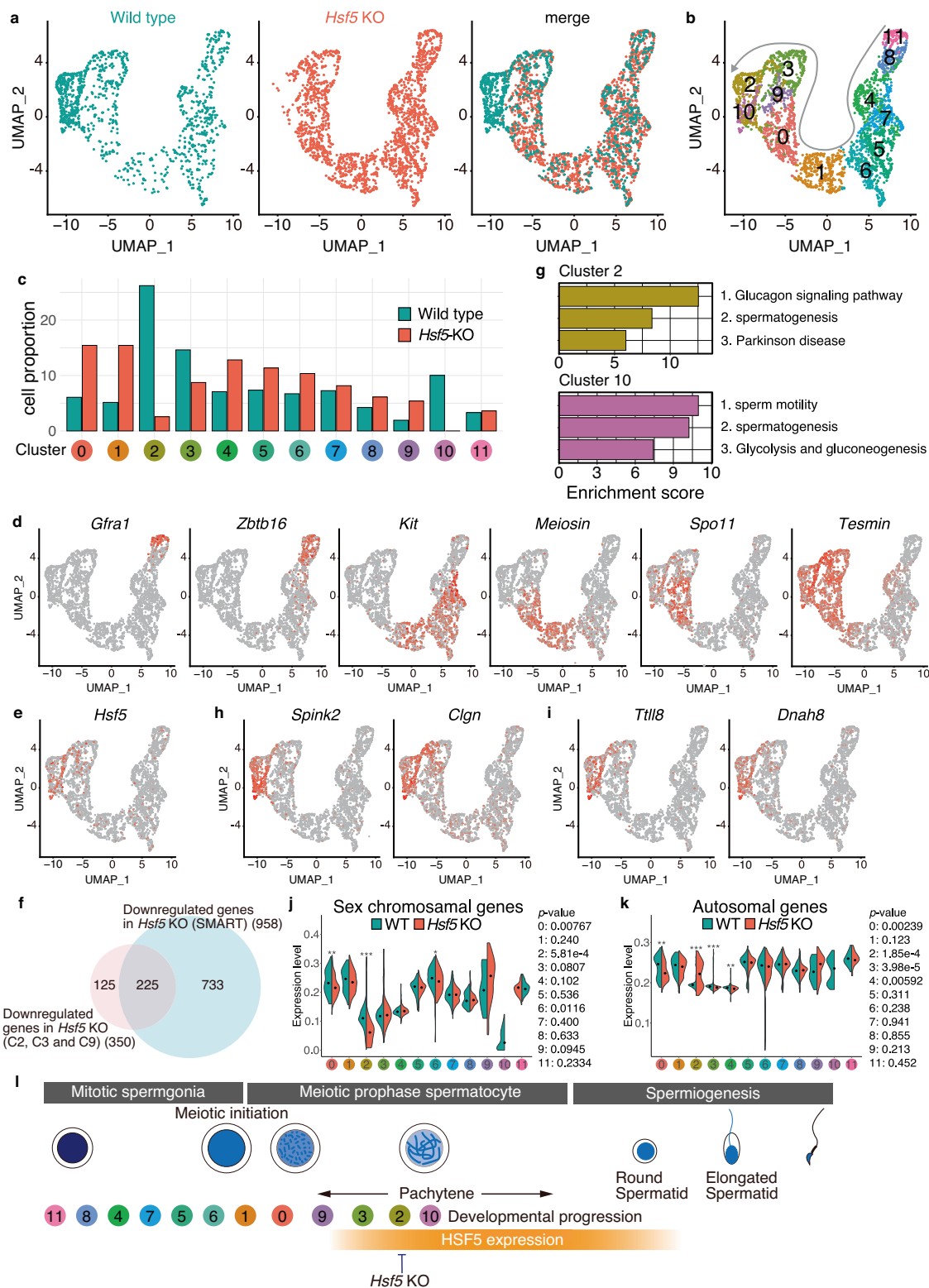

presumably corresponding to the most advanced stages of meiotic prophase in WT spermatocytes at P16. Intriguingly, Clusters 2 and 10 coincided with the abrupt upregulation of *Hsf5* expression in the WT cells (Fig. 6e). Thus, the subtype of spermatocytes that represented Cluster 2 was less frequently observed and that of Cluster 10 was absent in *Hsf5* KO spermatocytes, which were reciprocal to the upregulation of *Hsf5* in WT testes in Clusters 2 and 10. Furthermore, we cross-analyzed all the 958 downregulated genes identified in the SMART RNA-seq data (Supplementary Fig. 4d, e) with the Clusters in

the scRNA-seq data. This analysis revealed that 64% of the downregulated genes in *Hsf5* KO, which were identified in Clusters 2, 3, and 9 by scRNA-seq, were well overlapped with the downregulated genes in *Hsf5* KO identified by SMART RNA-seq (Fig. 6f). The overlap by these two different transcriptome analyzes suggested that HSF5 positively regulated expression of those overlapping genes in the subpopulations represented by Clusters 2, 3, and 9. Consistent with the cytological observation that *Hsf5* KO spermatocytes progressed through the early meiotic prophase but were eliminated by apoptosis at

**Fig. 6 | scRNA-seq analysis of WT and *Hsf5* KO spermatogenic germ cells.**
**a** UMAP representation of scRNA-seq transcriptome profiles for germ cells from P16 WT and *Hsf5* KO testes. **b** Clustering analysis of different gene expression patterns on UMAP-defined scRNA-seq transcriptomes of P16 WT and *Hsf5* KO cells. Gray arrow indicates developmental direction. **c** Bar graph showing the proportion of WT and *Hsf5* KO germ cells among the clusters. **d** UMAP plots show mRNA levels of key developmental marker genes of spermatogenic cells. Key developmental marker genes include *Gfra1*: spermatogonial stem cell, *Zbtb16*: undifferentiated spermatogonia, *Kit*: differentiating spermatogonia, *Meiosin*: pleleptotene spermatocyte, *Spo11*: early meiotic prophase spermatocyte, *Tesmin*: mid-pachytene spermatocyte. **e** The mRNA levels of *Hsf5* on the UMAP plot. **f** Venn diagram representing the overlap between the 958 downregulated genes identified in SMART RNA-seq data (Supplementary Fig. 4) and Downregulated genes in Clusters 2, 3, and 9 of Hsf5 KO. **g** Gene enrichment analysis of highly expressed genes in

Cluster 2 or Cluster 10. **h** Expression patterns of the representative genes in Cluster 2 (*Spink2, Clgn*) on the UMAP plot. **i** Expression patterns of the representative genes in Cluster 10 (*Ttll8, Dnah8*) on the UMAP plot. **j** The mRNA levels of the sex chromosomal genes among the clusters are shown in violin plots with a median. *p*-values by two-sided Wilcoxon rank sum test are shown on the right. **k** The mRNA levels of the autosomal genes among the clusters are shown in violin plots with a median. *p*-values by two-sided Wilcoxon rank sum test are shown on the right. **l** The subtype clusters delineated by scRNA-seq and the timing of HSF5 protein expressions are shown along the developmental stages. HSF5 started to appear in the spermatocyte nuclei from mid-pachytene onward, and were expressed in round spermatids. Vertical bars indicate the stages when the developmental progression is blocked in *Hsf5*KO spermatocytes. See also Supplementary Fig. 5, Supplementary Data 4. The schematic of developmental stages was adapted from our previously published paper (https://doi.org/10.1038/s41467-021-23378-4)[11].

mid-pachytene (Fig. 4), HSF5 was required for spermatocytes to progress beyond the substages of meiotic prophase that corresponded to Clusters 2 and 10.

Gene enrichment analysis revealed that genes related to spermatogenesis (*Spink2, Clgn, Hspa2, Ybx3, Tcp11, Rsph1, Pebp1, Ybx2, Nphp1, Catsperz, Ggnbp1, Psma8, Meig1, Ropn1l, Morn2*) were highly expressed in Clusters 2 and 10 (Fig. 6g, h, Supplementary Data 4). Furthermore, genes related to sperm motility, such as (*Ttll8, Dnah8, Ldhc, Sord, Ccdc39, Gk2, Cfap206, Zmynd10, Ropn1l, Spata33, Dnaaf1*) were highly expressed in Cluster 10 (Fig. 6g, i, Supplementary Data 4). Since *Spink2* and *Clgn* are known to be expressed in pachytene spermatocytes, Clusters 2 and 10 represent the subtypes of pachytene spermatocytes. In contrast, genes associated with Clusters 2 and 10 were underrepresented in *Hsf5* KO spermatocytes.

Notably, the overall expression level of sex chromosome genes exhibited a gradual downregulation in the progression from Cluster 2 to Cluster 10 (Fig. 6j). Thus, Clusters 2 and 10 represented the transitional stages where MSCI was in the process of being established. *Hsf5* KO spermatocytes exhibited downregulation of sex chromosome genes in Cluster 2. Therefore, we reasoned that *Hsf5* KO spermatocytes at least exhibited a sign of undergoing MSCI at this time point. Notably, overall expression level of autosomal genes was downregulated in the Cluster 3 subpopulation and then abruptly upregulated in Cluster 2 subpopulation of *Hsf5* KO (Fig. 6k), suggesting that some of, if not all, autosomal genes were derepressed in the absence of HSF5 in the subpopulation of Clusters 2. Thus, *Hsf5* KO spermatocytes exhibited gene expression changes at the mid-pachytene substage (Cluster 3) when the HSF5 protein should undergo an expression burst; consequently, specific subpopulations of mid-pachytene spermatocytes represented by Clusters 2 and 10 were lost in *Hsf5* KO testes (Fig. 6l). Therefore, HSF5 plays a role in regulating gene expression for the developmental progression beyond the mid-pachytene substage during spermatogenesis.

## HSF5 alters gene expression patterns by binding to promoter regions with a unique target specificity

HSF5 was predicted to possess a putative DNA-binding domain (Fig. 1b). We conducted CUT&Tag using spermatocytes enriched by fluorescent sorting from testes (Supplementary Fig. 6). CUT&Tag experiments were performed using three different antibodies against HSF5 (HSF5-N1, HSF5-N2, and HSF5-C). CUT&Tag analyzes revealed that HSF5 bound to 2191 sites (2087 genes) that were overlapped by the four-replicate dataset of HSF5 CUT&Tag (Fig. 7a). In addition, 86.4% of the peaks resided within 1 kb of the transcriptional start sites (TSSs) of the promoter region (Fig. 7a–c). The ATAC-seq analysis unveiled a modest elevation in chromatin accessibility at TSSs among HSF5-target genes in the *Hsf5* KO spermatocytes compared to the control (Supplementary Fig. 7), implying that HSF5 may contribute to regulation of chromatin organization at TSSs during the pachytene stage.

Since *Hsf5* KO spermatocytes were eliminated soon after they reached a specific point in the pachytene stage when the HSF5 protein should be upregulated (Fig. 6), it became technically challenging to compare the expression levels of HSF5-bound genes between WT and *Hsf5* KO spermatocytes at the specific time point when HSF5 is active in transcription. Alternatively, we analyzed the alteration of mRNA levels of HSF5-bound genes during spermatogenesis using the scRNA-seq data from spermatogenic cells (Fig. 6). Through hierarchical clustering, HSF5-bound genes were separated into three classes based on the alteration of mRNA levels across the order of the scRNA-seq Clusters that corresponded to the developmental direction (Fig. 7d, Supplementary Data 6). The genes assigned into Class 1 (476 genes) were upregulated in Clusters 2 and 10 upon *Hsf5* expression, and their functions were related to male gamete generation (such as *Piwil2* and *Kdm3a*), cilium organization, and intraflagellar transport. Those genes assigned to Class 2 (787 genes) were highly expressed in spermatogonia populations (Clusters 11, 8, 4, 7, 5, and 6 in scRNA-seq), and their functions were related to chromatin organization, RNA metabolism, and pluripotency. Those genes assigned to Class 3 (709 genes) were highly expressed in spermatocyte populations at earlier time points of meiotic prophase (Clusters 1, 0, and 9) and then abruptly downregulated upon *Hsf5* expression, and their functions were related to DNA metabolic process (such as *Atm, Dmc1, Sycp2, and Mybl1*), mRNA metabolic process, and protein localization to the organelle.

We further assessed the mRNA abundance of HSF5-target genes (Classes 1, 2, and 3) in the *Hsf5* KO. For this purpose, we reanalyzed the scRNA-seq dataset for the pachytene subpopulations (Clusters 9, 3, and 2). This allowed us to compare overall expression levels of HSF5-target genes between WT and *Hsf5* KO in the pachytene subpopulations, where *Hsf5* KO cells were still alive before apoptosis (Fig. 7e). On the one hand, Class 1 HSF5-target genes, which were upregulated upon *Hsf5* expression, were downregulated in *Hsf5* KO, suggesting that the Class 1 genes were activated by HSF5. On the other hand, Class 3 HSF5-target genes, which were downregulated upon *Hsf5* expression, were upregulated in *Hsf5* KO, suggesting that Class 3 genes were repressed by HSF5. Class 2 genes, which were highly expressed in spermatogonia populations, were upregulated in *Hsf5* KO. These observations suggest that HSF5 positively and negatively regulates the expression of HSF5-target genes to dynamically change the transcriptome in mid-pachytene spermatocytes.

## HSF5 possesses a unique target specificity

As another alternative confirmation, we further investigated HSF5-target sites in the genome using ChIP-seq analysis. HSF5 bound to 165 sites across the genome (161 nearest genes, which were assigned regardless of the distance from the HSF5-binding sites), of which 93.9% resided within 1 kb of the transcription start sites (TSS) in the promoter regions of the mouse genome (Supplementary Fig. 8a–c, Supplementary Data 5). Notably, 57.8% of the HSF5-bound genes revealed by ChIP-seq overlapped with those genes identified by CUT&Tag

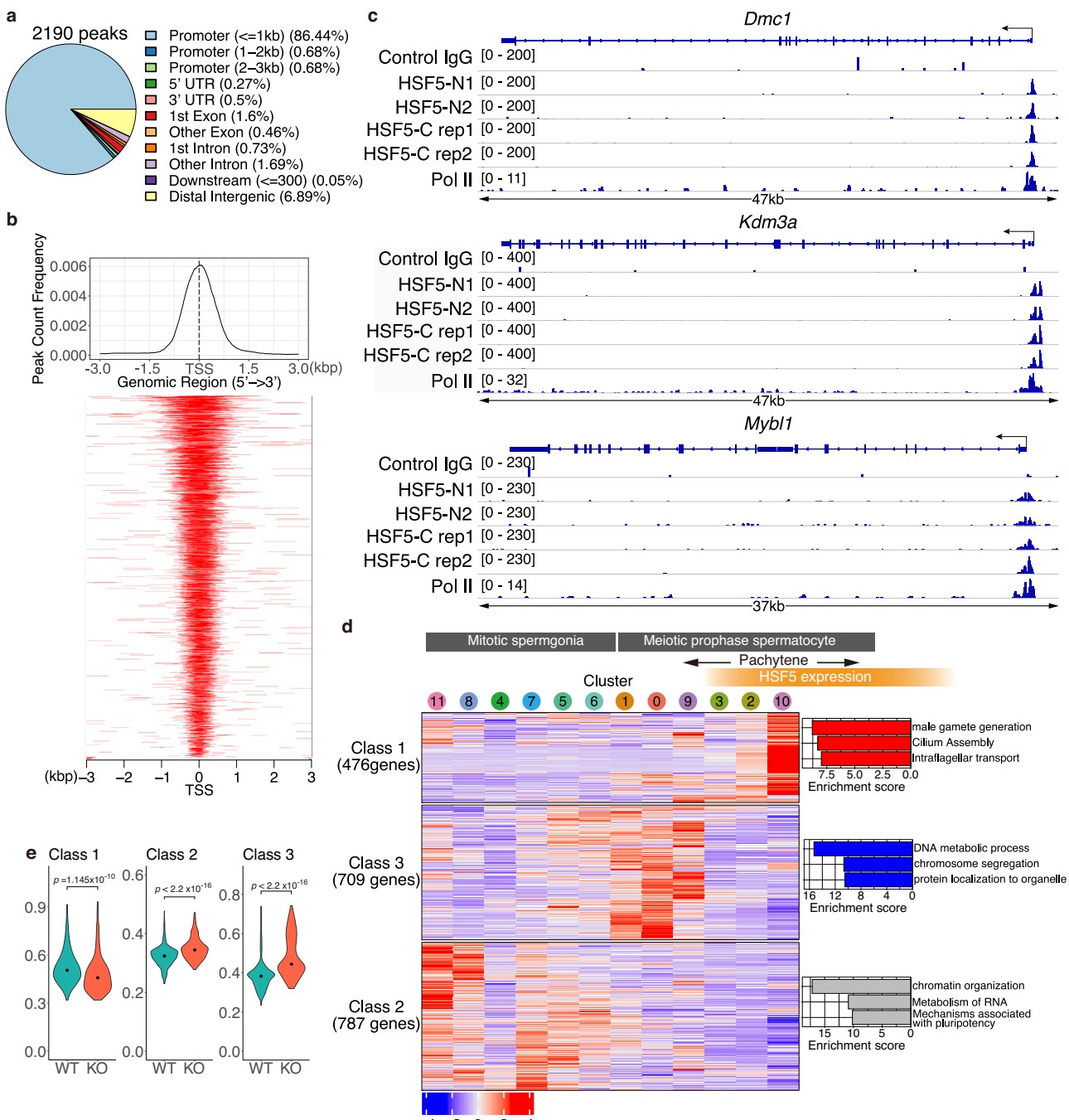

**Fig. 7 | HSF5 binds to the gene promoter regions. a** HSF5-bound peaks were commonly identified among the 4 replicates of HSF5 CUT&Tag (rabbit anti-HSF5-N1, rabbit anti-HSF5-N2, rabbit anti-HSF5-C antibodies), but not overlapped with control IgG CUT&Tag. HSF5 binding sites were classified by the genomic locations as indicated. **b** Heat map of the common HSF5 binding sites of HSF5 CUT&Tag at the positions −3.0 kb upstream to +3.0 kb downstream relative to the TSS. Average distributions of HSF5- ChIP-seq binding peak are shown on the top. **c** Genomic view of the CUT&Tag peaks revealed by HSF5-N1, HSF5-N2, HSF5-C (rep1, rep2) antibodies and control IgG over representative gene loci. Genomic coordinates were obtained from RefSeq. RefSeq IDs for mRNA isoforms are indicated. To specify testis specific transcription, binding peaks of RNA polymerase II in the testis are shown[48]. **d** Gene expression patterns of HSF5 bound genes during spermatogenesis were shown as Heatmap. Heatmaps show the hierarchical relationship of the expression patterns of the HSF5-bound genes across the developmental direction. Expression pattern of the HSF5-bound genes was assessed by scRNA-seq data of

spermatogenic cells as described in Fig. 6k. The cluster number is indicated on the top. The order of clusters from left to right corresponds to developmental direction of spermatogenesis (undifferentiated spermatogonia to pachytene spermatocyte). The expression patterns of the HSF5-bound genes were classified (Class1-Class3). On the right, top3 gene enrichment terms ranked by Enrichment score, log(q-value), are shown. **e** The average expression levels of the HSF5-target genes (Class 1, 2, and 3) are shown in violin plots with a median. The expression levels were compared between WT and *Hsf5* KO mice using the scRNA-seq data for pachytene sub-populations that were pooled from clusters 9, 3, and 2. Note that scRNA-seq data of cluster 10 was excluded since the cluster 10 sub-population was already eliminated in *Hsf5* KO mice. Statistical significance is shown by *p*-value (Class1; $p = 1.145 \times 10^{-10}$, Class 2; $p < 2.2 \times 10^{-16}$, Class 3; $p < 2.2 \times 10^{-16}$, two-sided Wilcoxon rank sum test). See also Supplementary Data 6 for a complete list of the HSF5-bound genes and the Gene enrichment analyzes.

(Supplementary Fig. 8d, Supplementary Data 6). Although the number of HSF5-bound genes revealed by ChIP-seq was fewer than those identified by CUT&Tag (Fig. 7), the overlapping genes from these two analyzes more strictly represent bona fide HSF5-bound genes.

DNA-binding motif analysis identified de novo HSF5-ChIP enriched DNA sequences that were different from the previously known binding motifs of other HSF family transcription factors (Supplementary Fig. 8e). While the typical HSE that resides in the promoter regions of heat shock-responsive genes is composed of three contiguous inverted repeats, nTTCnnGAAnnTTCn[45], the most probable HSF5-binding motif is composed of the sequence with a strong preference for a single triplet GAA (Supplementary Fig. 8d). It should be noted that HSF5-target genes were little overlapped with either of HSF1- target genes[46] or HSF2-target genes in the testis[47] (Supplementary Fig. 8f, Supplementary Data 5, 6). Thus, HSF5 binds to different target genes to those bound by HSF1 or HSF2 during spermatogenesis.

This suggests that HSF5 directly binds to promoters through at least, if not all, this motif, which is distinct from HSE bound by other canonical HSF family transcription factors.

To validate the DNA-binding specificity of HSF5 to the motif predicted from the HSF5 ChIP-seq data, we examined the DNA-binding ability of the HSF5 DNA-binding domain (HSF5-DBD) using an in vitro EMSA (Fig. 8a). Indeed, HSF5-DBD bound to DNA with the target motif but not DNA with mutant sequences (Fig. 8b). As the probe contained three octamer motif units, three shifted bands appeared when increasing amounts of HSF5-DBD were added. In contrast, all mobility shifts of the bands were completely abolished when substitutions were introduced into the octamer motif. We confirmed that the HSF5 protein-DNA complex was titrated away in the presence of an excess amount of unlabeled DNA with the target motif but not the mutant sequence, showing the DNA-binding specificity of HSF5 to the predicted motif (Fig. 8c). Thus, the predicted motif with a single triplet GAA was sufficient for the DNA binding of HSF5-DBD, though we do not exclude a possibility that HSF5-DBD recognizes other sequences than this motif.

To elucidate how HSF5 is involved in transcriptional regulation, factors interacting with HSF5 were screened by immunoprecipitation (IP) using different HSF5 antibodies followed by mass spectrometry (MS). Our HSF5 IP-MS analysis demonstrated that HSF5 immunoprecipitated with the subunits of the SWI/SNF chromatin-remodeling complex, SMARCA5, SMARCC2, SMARCE1 and SMARCC1, although a low number of peptides was identified (Supplementary Fig. 9a, b, Supplementary Data 7). However, further validation by HSF5 IP-Western Blot (WB) showed that SMARCA5, a SWI/SNF subunit, was below the detection threshold for western blotting (Supplementary Fig. 6c). Thus, the HSF5-SWI/SNF chromatin-remodeling complex interaction could be transient or regulatory rather than stoichiometric. Similarly, HSF5 was repeatedly immunoprecipitated with KCTD19 and HDAC1 (Supplementary Fig. 9a, b, Supplementary Data 7). Our previous study demonstrated that KCTD19 reciprocally co-immunoprecipitates with HSF5 in chromatin extracts[11]. KCTD19 forms a subcomplex that consists of KCTD19, HDAC1/2, TRERF1, and TDIF1, and interacts with DNA-binding proteins such as ZFP541 and MIDEAS[11–13,48], thus acting as a corepressor of transcriptional repression during spermatogenesis. However, further validation by KCTD19 IP-WB showed that HSF5 was below the detection threshold for western blotting (Supplementary Fig. 9c). Although it is possible that HSF5 represses the target genes collaborating with the KCTD19-HDAC1/2-containing complex, the HSF5-KCTD19 interaction could be transient or regulatory rather than stoichiometric. The precise molecular mechanism by which HSF5 operates in transcriptional activation and repression awaits further investigation.

## Discussion

### HSF5 plays a specific role in the developmental progression of spermatocytes under non-stress conditions

Our genetic and cytological analyzes demonstrated that HSF5 is essential for progression beyond the pachytene stage under non-stress conditions during spermatogenesis, rather than being specifically associated with the heat stress response, which is consistent with previous work[33]. Chromatin binding analysis combined with transcriptome analyzes revealed that HSF5 is required for the suppression of pre-meiotic and meiotic genes, as well as the upregulation of post-meiotic genes. HSF5 recognizes a DNA motif different from the typical heat shock elements recognized by other canonical HSFs (Fig. 8). Thus, HSF5 stands out as an atypical HSF due to its target specificity and requirement for developmental regulation rather than being primarily associated with the heat stress response.

While partially agreeing with previous findings[33], our study reveals several differences in the observations and interpretations of the data compared to the previous study on *Hsf5* KO mice[33]. Previously, HSF5 appeared at the pachytene stage, predominantly localizing to the XY body in spermatocytes. In *Hsf5* KO pachytene spermatocytes, elongated XY chromosomes were observed instead of the typical condensed XY body, leading to the failure of MSCI and an upregulation of sex chromosomal genes, as revealed by RNA-seq data[33]. In this study, we observed that HSF5 localizes across the nuclei of pachytene spermatocytes, with a particular concentration at the XY body, especially in stage VII–VIII pachytene spermatocytes (Fig. 2a, d, Supplementary Fig. 2a–c). However, most *Hsf5* KO pachytene spermatocytes exhibited a typical XY body that was accompanied by γH2AX signals and the exclusion of RNA pol II (Fig. 4c), while the remaining ~24.2% showed dispersed γH2AX signals (Fig. 4d). Moreover, we did not see an upregulation of sex chromosomal genes in our SMART-seq data (Supplementary Fig. 4) or our scRNA-seq data in *Hsf5* KO pachytene spermatocytes (Fig. 6j). Thus, our interpretation is that some, if not all, *Hsf5* KO pachytene spermatocytes form the XY body and establish MSCI. Furthermore, while the previous study showed that SMARCA4, a subunit of the SWI/SNF chromatin-remodeling complex, was downregulated in *Hsf5* KO pachytene spermatocytes[33], we did not observe a downregulation of *Smarca4* or any of other SWI/SNF components in our SMART-seq data.

Although we do not know the exact reason for the different observations between the previous study and ours, one explanation could be the use of different methodologies for spermatocyte isolation: fluorescent sorting of DCV-stained spermatocytes from P17 testes (this study) versus spermatocyte isolation from testes of unknown age using STA-PUT[33]. This discrepancy in isolation methods may have caused differences in cellular populations between the WT and *Hsf5* KO samples.

### HSF5 plays a distinct role to other HSF family in the testes

In mouse testes, *Hsf5* and other HSF family members, *Hsf1, Hsf2*, and *Hsfy2*, are expressed at different or overlapping stages of spermatogenesis (Supplementary Fig. 1b)[49]. The testis is a heat-sensitive organ where spermatogenesis occurs under low-temperature conditions[50,51]. In mice, when the testes experience temperature elevation, spermatogenesis is compromised during meiotic recombination, leading to the elimination of spermatocytes[52]. A previous study showed that transgenic mice constitutively overexpressing an active form of HSF1 in the testes are infertile due to a block in spermatogenesis and apoptosis, whereas female fertility is unaffected[53–55]. Thus, HSF1 acts, at least in part, as a stress response factor in the cell-death decision at the pachytene stage under the stress condition in the testes. In contrast, HSF5 plays a specific role in developmental progression under non-stress conditions rather than heat stress response in mice (Figs. 3, 4, 5).

It has been shown that HSF1 and HSF2 play a role in spermatogenesis in non-stress conditions. *Hsf1* KO[27] and *Hsf2* KO[28,29] males

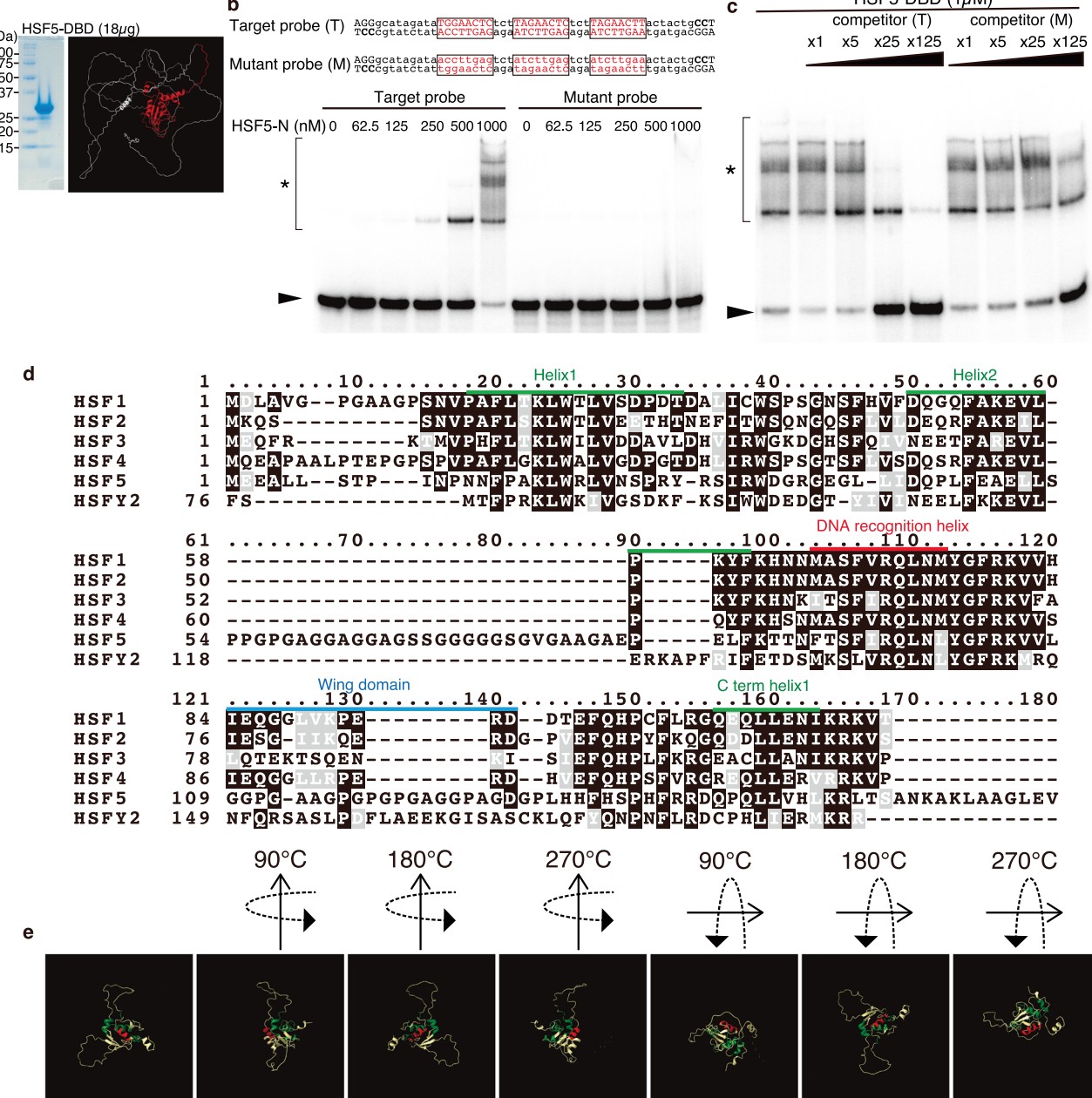

**Fig. 8 | HSF5 possesses an unique DNA binding specificity. a** The purified HSF5 N-terminal (aa1-209) protein was used for EMSA assay (left). Shown on the right is the corresponding HSF5 N-terminal part is shown (red) on the ribbon model that was predicted from AlphaFold2. **b** DNA binding ability of HSF5 DNA binding domain was examined by EMSA assay. Shown on the top are the target (T) and mutant (M) sequences of the DNA probes. Target (T) sequence was designed according to the enriched motif that was predicted by Chip-seq as shown in Fig. S8e. Increasing amount of the purified protein was mixed with 0.04 pmol of [32]P-labeled DNA probes (T or M) at the protein/DNA molar ratio of 15.6 – 250. Arrowhead: unbound DNA. The protein-DNA complexes are shown by * with a bracket. **c** HSF5 N-terminal protein (1 μM) was mixed with 0.04 pmol of the [32]P-labeled target (T) DNA probe. DNA binding specificity of HSF5- DNA complex was assessed by adding the unlabeled target or mutant competitor DNA (1 – 125 fold excess to the [32]P-labeled DNA probes). A single experiment was performed in (**a**–**c**). **d** Amino acid sequences of the DNA-binding domain in the mouse HSFs are aligned. HSF1, HSF2, and HSF4 possess a DNA recognition helix (red) containing a conserved Ser-Phe-Val-Arg-Gln amino acid sequence, which is known to insert into the major groove of the HSE. HSF5 possesses a Ser-Phe-Ile-Arg-Gln amino acid sequence at the corresponding position. HSF5 possesses insertion of amino acid sequence between the helix 2 and DNA recognition helix. **e** Ribbon models of HSF5-DBD (1-167 a.a.) are shown that were predicted from AlphaFold2. Helixes are colored as shown in (**a**).

exhibit reduced spermatogenesis but are fertile. In *Hsf1* and *Hsf2* double KO males, spermatocytes fail to progress to the pachytene stage, leading to a complete lack of mature sperm, resulting in male infertility[30]. HSF1 and HSF2 regulate the expression of the common target genes on the sex chromosomes in spermatogenic cells[46,47]. Thus, HSF1 and HSF2 have complementary or overlapping roles in meiotic prophase progression, as suggested by their ability to

heterodimerize[56]. Similarly, we showed that *Hsf5* KO spermatocytes failed to progress beyond the pachytene stage and were consequently eliminated by apoptosis (Fig. 4g). Whereas HSF1 faintly appears in the nuclei of early pachytene spermatocytes of stage I, localizing to the sex body in pachytene spermatocytes and the chromocenter in round spermatids[46], HSF5 appears in the nuclei of mid pachytene spermatocyte localizing to the sex body particularly at the stage VII-VIII, and in

the round spermatids with rather excluded from the chromocenter (Fig. 2a, d, Supplementary Fig. 2a, b), indicating the spatially and temporally different behaviors of HSF 1 and HSF5. Crucially, these evidences suggest that HSF5 has a specific function in meiotic prophase progression that cannot be compensated for by HSF1 or HSF2 and vice versa. Therefore, HSF5 plays a distinct role in spermatogenesis, different from HSF1 and HSF2 in mice. Consistently, in zebrafish, *Hsf5* mutant males are infertile with arrest at the zygotene-pachytene transition but still show a normal heat stress response[31]. Therefore, the role of HSF5 in the progression of spermatogenesis under non-stress conditions is evolutionarily conserved in mammals and fish.

HSFY has been implicated in male infertility in humans[57]. Although *Hsfy2* is expressed in spermatogenic cells in the mouse testes (Supplementary Fig. 1b), it is unknown whether mouse HSFY2 is involved in spermatogenesis. Nevertheless, it is unlikely that HSFY2 has an overlapping function that genetically complements the role of HSF5 in *Hsf5* KO pachytene spermatocytes.

### HSF5 acts as an atypical HSF transcription factor

HSF5 differs from other canonical HSFs (HSF1, HSF2, and HSF4) in terms of protein structure (Fig. 1b), expression pattern (Fig. 1d), DNA-binding specificity (Figs. 7, 8, Supplementary Fig. 8), and physiological function (Figs. 3–6). HSF family members are classified as transcription factors that possess an evolutionarily conserved winged-helix-turn-helix (wHTH) DNA-binding domain in fungi, invertebrates, and vertebrates, and were originally described to recognize a consensus HSE[23].

While the typical HSE is composed of three contiguous inverted repeats of nGAAn; (nGAAnnTTCnnGAAn) or its complementary sequence (nTTCnnGAAnnTTCn)[23,45], our ChIP-seq data predicted that HSF5-binding motif is composed of a single triplet GAA (Supplementary Fig. 8e). The DBDs of HSF1, HSF2, and HSF4 possess a recognition helix that contains a conserved Ser-Phe-Val-Arg-Gln amino acid sequence inserted into the major groove of the HSE, in which a conserved Arg residue forms hydrogen bonds with the guanine of GAA and is essential for DNA binding of the HSF[58]. In contrast, HSF5 possesses a Ser-Phe-Ile-Arg-Gln amino acid sequence at the same position (Fig. 8d, e), in which Ile, with its bulky side chain, was placed instead of Val at the neighboring position of the essential Arg. Strikingly, HSF5 possesses an insertion of an amino acid sequence between helix 2 and the DNA recognition helix, which corresponds to the predicted IDR (Fig. 1b). The wing domain is poorly conserved in HSF5; however, it appears to have been substituted with a predicted IDR (Fig. 1b). Furthermore, whereas HSF1, HSF2, and HSF4 possess two heptad repeats, HR-A and HR-B, that are predicted to form inter-molecular leucine zippers for homotrimer oligomerization[34], HSF5 lacks these heptad repeats (Fig. 1b). It is possible that because of these structural differences, HSF5 acquires target specificity that is distinct from that of HSF1, HSF2, and HSF4. Indeed, HSF2 binds to and regulates genes on the Y chromosome long arm (MSYq), such as *Sly and Ssty2* in spermatogenic cells[47]. HSF1 also binds to the multi-copy genes on the sex chromosomes that are shared by HSF2, and other genes[46]. HSF5 commonly bound to only a minor fraction of HSF1- and HSF2-target genes (Supplementary Fig. 8f, Supplementary Data 5, 6). These evidences highlight the different target specificities of HSF1, HSF2 and HSF5 during spermatogenesis. Since HSF5 is prone to bind to promoter regions with preference for GAA sequence, it is possible that the promoter binding of HSF5 is tuned by cooperation with other DNA-binding factors (Supplementary Fig. 9). Altogether, HSF5 acts as an atypical HSF transcription factor so that it executes a more pronounced role in regulating developmental genes rather than stress response genes through different mechanisms of DNA binding from canonical HSFs.

### HSF5 plays an essential role in pachytene progression during spermatogenesis

In males, meiotic prophase is accompanied by significant changes in gene expression programs[59–63]. At the pachytene stage in males, the progression of meiotic prophase is monitored under several layers of regulation, such as the pachytene checkpoint[64] and MSCI[41,65,66]. Concurrent with surveillance mechanisms, multiple gene regulatory programs are imposed on the male meiotic prophase to circumvent the barrier at the pachytene stage[5–8].

Our study revealed that HSF5 is a spermatocyte-specific transcription factor essential for progression beyond the pachytene stage. HSF5 positively and negatively regulates the expression of HSF5-target genes to dynamically change the gene expression network in mid-pachytene spermatocytes (Fig. 7).

Our IP-MS analysis of the HSF5 immunoprecipitates detected KCTD19 (Supplementary Fig. 9), which has been shown to form a ZFP541-HDAC1/2-containing repressive complex[11], and we reasoned that HSF5 plays a role at least in repressing these target genes during pachytene progression. This is consistent with previous studies showing that pre-pachytene gene expression programs are suppressed by the ZFP541-KCTD19-containing repressive complex at the pachytene exit[11–13,48].

In addition, HSF5 may positively regulate gene expression in cooperation with other transcription factors. Presumably, the absence of HSF5 indirectly delays DSB repair processes at the pachytene stage as a secondary effect of aberrant gene expression (Fig. 4). Therefore, HSF5 may trigger the reconstruction of the transcription network to promote pachytene progression and facilitate spermatid production. Our study sheds light on the regulatory mechanisms of gene expression that promote the developmental progression of the meiotic prophase, leading to spermatid differentiation.

## Methods

### Animals

*Hsf5* knockout (*Hsf5* KO) mice were with the C57BL/6 background. *Hsf5-3xFLAG-HA* knock-in mice were C57BL/6 background (age: 5 weeks old). Male mice were used for ChIP-seq (age: P10-21), CUT&Tag (age: P18-19), immunoprecipitation of testis extracts (age: P18-23, 4 weeks and 8 weeks old), histological analysis of testes, and immunostaining of testes, RNA extraction (age: P15-21, 4 weeks and 8 weeks old). Female mice were used for histological analysis of the ovaries and RNA extraction (age: 4 weeks and 8 weeks old) (E12.5-18.5). Whenever possible, each knockout animal was compared among littermates or age-matched non-littermates from the same colony, unless otherwise described. Housing conditions for the mice were under 12 h dark/12 h light cycle, ambient temperature at 20–23 °C and humidity 40–60%. Animal experiments were approved by the Institutional Animal Care and Use Committee of Kumamoto University (approval F28-078, A2022-001).

### Generation of *Hsf5* knockout mice and genotyping

*Hsf5* knockout mice were generated by introducing Cas9 protein (317-08441; NIPPON GENE, Toyama, Japan), tracrRNA (GE-002; FASMAC, Kanagawa, Japan), synthetic crRNA (FASMAC), and ssODN into C57BL/6 N fertilized eggs using electroporation. For the generation of *Hsf5* Exon1-6 deletion (Ex1-6Δ) allele, the synthetic crRNAs were designed to direct CCTTAAATTCAAATTAGATG(AGG) of the 5'upstream of *Hsf5* exon 1 and ATGTAGACAAAAGCACTGAG(AGG) in the exon 6. ssODN: 5'- AAAAATCTAAAATAAGAAAACAGTGTTAACCTCTCATGAGGTCCTT GGTACCTGGCAGAAGGGAATAAAG −3' was used as a homologous recombination template.

The electroporation solutions contained tracrRNA(10 μM), synthetic crRNA(10 μM), Cas9 protein (0.1 μg/μl), ssODN (1 μg/μl) for *Hsf5* knockout in Opti-MEM I Reduced Serum Medium (31985062; Thermo Fisher Scientific). Electroporation was carried out using the Super

Electroporator NEPA 21 (NEPA GENE, Chiba, Japan) on Glass Microslides with round wire electrodes, 1.0 mm gap (45-0104; BTX, Holliston, MA). Four steps of square pulses were applied (1, three times of 3 mS poring pulses with 97 mS intervals at 30 V; 2, three times of 3 mS polarity-changed poring pulses with 97 mS intervals at 30 V; 3, five times of 50 mS transfer pulses with 50 mS intervals at 4 V with 40% decay of voltage per each pulse; 4, five times of 50 mS polarity-changed transfer pulses with 50 mS intervals at 4 V with 40% decay of voltage per each pulse).

The targeted *Hsf5* Exon1-6Δ allele in F0 mice were identified by PCR using the following primers; Hsf5-g-F1: 5′-GGGAGATCATAGC TGGTCATTAAGC-3′ and Hsf5-g-R2: 5′- CAGAGGGATAAGAAAA TTGGTGATAG-3′ for the knockout allele (300 bp). Hsf5-g-F1 and Hsf5-g-R1: 5′-TCTCCCACCGTTCTCGATCC-3′ for the Ex1 of WT allele (678 bp). The PCR amplicons were verified by Sanger sequencing. Three lines of KO mice (#5, #9, #21) were established. Line #21 of *Hsf5* KO mice was used in most of the experiments, unless otherwise stated. Primer sequences are listed in Supplementary Data 1.

### Generation of *Hsf5-3xFLAG-HA* knock-in mice and genotyping

*Hsf5-3xFLAG-HA* knock-in mouse was generated by introducing Cas9 protein, tracrRNA, synthetic crRNA, and ssODN into C57BL/6 N fertilized eggs using electroporation as described above. The synthetic crRNA was designed to direct GAGTTAAAAGAATGAGAAGC(TGG) of the *Hsf5*.

ssODN: 5′-ACGTGGCCTGCAAGCAGGAACACTTCCCAAAGGAGGA GGAGTTAAAAGAAGGAGACTACAAAGACCATGACGGTGATTATAAAG ATCATGACATCGATTACAAGGATGACGATGACAAGGGATACCCCTAC GACGTGCCCGACTACGCCTAAGAAGCTCCGTGACAGCCGTGACATCG TGCGCTAGCCACAGCTGGAGGGGA-3′ was used as a homologous recombination template.

The targeted *Hsf5-3xFLAG-HA* knock-in allele in F0 mice was identified by PCR using the following primers: Hsf5-3xFLAG-HA_genotyping_3F: 5′- TGAAGGCATGTCTGTTGAC GTC −3′ and Hsf5-3xFLAG-HA_genotyping_1R: 5′- GCATCACGACTCA GCACACA-3′ for the knock-in allele (598 bp), and for the wild-type allele (499 bp). The PCR amplicons were verified by sequencing. Primer sequences are listed in Supplementary Data 1.

### Histological analysis

For hematoxylin and eosin staining, testes, epididymis and ovaries were fixed in 10% formalin or Bouin solution and embedded in paraffin. Sections were prepared on CREST-coated slides (Matsunami) at 6 μm thickness. The slides were dehydrated and stained with hematoxylin and eosin.

For Immunofluorescence staining, testes were embedded in Tissue-Tek O.C.T. compound (Sakura Finetek) and frozen. Cryosections were prepared on the CREST-coated slides (Matsunami) at 8 μm thickness and then air-dried and fixed in 4% paraformaldehyde in PBS at pH 7.4. The serial sections of frozen testes were fixed in 4% PFA for 5 min at room temperature and permeabilized in 0.1% TritonX100 in PBS for 10 min. The sections were blocked in 3% BSA/PBS or Blocking One (Nakarai), and incubated at room temperature with the primary antibodies in a blocking solution. After three washes in PBS, the sections were incubated for 1 h at room temperature with Alexa-dye-conjugated secondary antibodies in a blocking solution. PNA lectin staining was performed using Lectin from *Arachis hypogaea*, FITC conjugate (1:1000, Sigma-Aldrich L7381). TUNEL assay was performed using MEBSTAIN Apoptosis TUNEL Kit Direct (MBL 8445). DNA was counterstained with Vectashield mounting medium containing DAPI (Vector Laboratory). Statistical analyzes, and production of graphs and plots were done using GraphPad Prism9 or Microsoft Excel.

### Immunostaining of spermatocytes

Surface-spread nuclei from spermatocytes were prepared by the dry down method as described[67,68] with modification. The slides were then air-dried and washed with water containing 0.1 % TritonX100 or frozen for longer storage at −30 °C. The slides were permeabilized in 0.1% TritonX100 in PBS for 5 min, blocked in 3% BSA/PBS, and incubated at room temperature with the primary antibodies in 3% BSA/PBS. After three washes in PBS, the sections were incubated for 1 h at room temperature with Alexa-dye-conjugated secondary antibodies (1:1000; Invitrogen) in a blocking solution. For immunostaining of FACS-isolated spermatocytes, cells were suspended in PBS without hypotonic treatment and structurally preserved nuclei of spermatocytes were prepared by cytospin at 1000 rpm for 5 min (Thermofisher). Cells were fixed with 4% PFA in PBS for 5 min. The slide grasses were washed with PBS containing 0.1% Triton-X100 in PBS. After washing with PBS, immunofluorescence staining was performed immediately. DNA was counterstained with Vectashield mounting medium containing DAPI (Vector Laboratory).

### Whole-mount immunostaining of seminiferous tubules

GFP-positive seminiferous tubules were excised from *Stra8-3xFLAG-HA-p2A-GFP* knock-in mouse under OLYMPUS SZX16 fluorescence stereo microscope, fixed in 4% paraformaldehyde in PBS overnight at 4 °C, and washed with 0.1% Triton X100/PBS for 1 h at room temperature. Blocking was done with 2.5% normal donkey serum/2.5% normal goat serum/0.1% Triton X100/PBS for 1 h at room temperature. Immunostaining was done as described above.

### Immunostaining of squashed spermatocytes

GFP-positive seminiferous tubules were excised from *Stra8-3xFLAG-HA-p2A-GFP* knock-in mouse under OLYMPUS SZX16 fluorescence stereo microscope, fixed in 4% paraformaldehyde in PBS for 5 min at room temperature, and washed with PBS for 5 min. Tubule bunches were then put on glass slides with 50 μL of PBS. Pressure was applied on a coverslip to disperse cells from tubules, followed by freezing at −80 °C with the coverslip for longer storage. Blocking was done with 3% BSA in PBS for 1 h at room temperature. Immunostaining was done as described above.

### Imaging

Immunostaining images were captured with DeltaVision (Cytiva). The projection of the images was processed with the SoftWorx software program (Cytiva). All images shown were Z-stacked. Excitation intensity and exposure time were adjusted for each condition, and images were simultaneously acquired at the same condition for comparable analyzes. The brightness of images was linearly adjusted to the same range of scale on the signal intensity histogram using SoftWorx software program for better visibility. For counting seminiferous tubules, immunostaining images were captured with BIOREVO BZ-X710 (KEYENCE), and processed with BZ-H3A program. XY-stitching capture by 10x objective lens was performed for multiple-point color images using BZ-X Wide Image Viewer. Images were merged over the field using BZ-H3A Analyzer (KEYENCE). Immunostaining images of whole mount seminiferous tubules were captured with confocal microscope TSC SP8 and processed with LASX ver.2 Software (Leica). Fluorescent and bright field images were captured with OLYMPUS BX53 fluorescence microscope and processed with CellSens standard program.

### Production of antibodies against HSF5

Polyclonal antibodies against mouse HSF5 N-terminal (aa1-209) and HSF5 C-terminal (aa324-624) were generated by immunizing rabbits. His-tagged recombinant proteins of HSF5 N-terminal region (aa1-209) and HSF5 C-terminal region (aa324-624) were produced by inserting cDNA fragments in-frame with pET19b (Novagen) respectively in *E. coli*

strain BL21-CodonPlus (DE3)-RIPL (Agilent), solubilized in a denaturing buffer (6 M HCl-Guanidine, 20 mM Tris-HCl pH 7.5) and purified by Ni-NTA (QIAGEN) under denaturing conditions. The antibodies were affinity-purified from the immunized serum with immobilized antigen peptides on CNBr-activated Sepharose (GE Healthcare).

## Antibodies

The following antibodies were used for immunoblot (IB) and immunofluorescence (IF) studies: Guinea pig anti-SYCP3 (IF, 1:2000, our home made)[16], Rat anti-SYCP3 (IF, 1:1000, our home made)[16], Rabbit anti-SYCP1 (IF, 1:1000, Abcam ab15090), Mouse anti-γH2AX (IF, 1:1000, Abcam ab26350), Mouse anti-γH2AX (IF, 1:1000, Merck Millipore 05-636), Rabbit anti-γH2AX (IF, 1:1000, Abcam ab11174), Rat anti-STRA8 (IF, 1:1000, our homemade)[16], Guinea pig anti-H1t (IF, 1:2000, kindly provided by Marry Ann Handel)[38], Rabbit anti-α-tubulin DM1A (IB, 1:2000, Sigma 05-829), Rabbit anti-HSF5 N (IB,IF, 1:1000, our home made)(this paper), Rabbit anti-HSF5 C (IB,IF, 1:1000, our home made)(this paper), Mouse anti-FLAG M2 (IB, 1:1000, Sigma-Aldrich F1804), Rabbit anti-HA (IB,IF, 1:1000, Abcam ab9110), Mouse anti-MLH1 (IF, 1:200, Cell Signaling 3515), Rabbit anti-DMC1 (IF, 1:500, Santa Cruz SC-22768), Mouse anti-RNA Pol II (8WG16) (IF, 1:200, Santa Cruz: SC-56767), Rabbit anti-BRCA1 (IF, 1:2000, kindly provided by Satoshi Namekawa)[20], Rabbit anti-KCTD19-N (IB, 1:1000, our home made)[11], Rabbit anti-HDAC1(IB, 1:1000, Abcam ab19845), Rabbit anti-TDIF1(IB, 1:1000, Abcam ab228703), Rabbit anti-SNF2H/SMARCA5 (IF, 1:1000, Abcam ab72499), Rabbit anti-Actin (IB, 1:1000, Sigma A2066).

Following secondary antibodies were used: Goat anti-Rabbit IgG Alexa 488 (IF, 1:1000, Invitrogen 31570), Goat anti-Gunia pig IgG-Alexa Fluour 488 (IF, 1:1000, Abcam ab150185), Donkey Anti-Mouse IgG Alexa 555 (IF, 1:1000, Invitrogen A31570), Goat Anti-Rat IgG Alexa 555 (IF, 1:1000, Thermo Fisher A21434), Goat anti-Gunia pig IgG-Alexa Fluour 555 (IF, 1:1000, Abcam ab150186), Goat anti-rabbit IgG-Alexa Fluour 647 (IF, 1:1000, Thermo Fisher, A21244), Donkey anti-mouse IgG-Alexa Fluour 647 (IF, 1:1000, Thermo Fisher, A31571), Sheep anti-mouse IgG-Horseradish Peroxidase (IB, 1:10000, Cytiva, NA931), Donkey anti-rabbit IgG-Horseradish Peroxidase (IB, 1:10000, Cytiva, NA934).

## PCR with reverse transcription

Total RNA was isolated from tissues and embryonic gonads using TRIzol (Thermo Fisher). cDNA was generated from total RNA using Superscript III (Thermo Fisher) followed by PCR amplification using Ex-Taq polymerase (Takara) and template cDNA. For RT-qPCR, total RNA was isolated from WT embryonic ovaries (E12.5 – E18.5), and WT testes, and cDNA was generated as described previously[11]. *Hsf5* cDNA was quantified by DCT method using TB Green Premix Ex Taq II (Tli RNaseH Plus) and Thermal cycler Dice (Takara), and normalized by *GAPDH* expression level.

For RT-qPCR of HSR genes after heat shock, the seminiferous tubules from whole testes of male *Hsf5* +/- (n = 3) and KO (n = 3) mice at P16 were unraveled and incubated in DMEM containing 10% FBS at 33 and 37 °C for 3 h. For RT-qPCR, total RNAs were prepared by RNeasy Mini Kit (QUIAGEN, 74104). *Hsf5*, *Hspa1a,* and *Hsfpa1a* cDNAs were quantified as described above.

Sequences of primers used for RT-PCR and RT-qPCR were as follows:

Hsf1_RTPCR_F1: 5′-GCCCCTCTTCCTTTCTGCAT-3′
Hsf1_RTPCR_R1: 5′-TCATGTCGGGCATGGTCAC-3′
Hsf2_RTPCR_F1: 5′-TCCTGTTAGCAGAAACGGCA-3′
Hsf2_RTPCR_R1: 5′-GGGATCACTTCCAAAGACGA-3′
Hsf3_RTPCR_F1: 5′-AGCTTGATCTCAGTGGGGGA-3′
Hsf3_RTPCR_R1: 5′-ACTAGCCAGCAGCCATTGAA-3′
Hsf4_RTPCR_F1: 5′-GAACTCAGGCAGCAGAACGA-3′
Hsf4_RTPCR_R1: 5′-GGAGGGGCGACTGGATAAAG-3′
Hsfy2_RTset_1F: 5′-AATGCAGGCTGTTTCCCCTA-3′

Hsfy2_RTset_1R: 5′-GGTATGCGGTGGCCTCTTTA-3′
Gapdh_F2(Gaphdh5599): 5′-ACCACAGTCCATGCCATCAC-3′
Gapdh_R2(Gapdh5600): 5′-TCCACCACCCTGTTGCTGTA-3′
Gapdh_F: 5′-TTCACCACCATGGAGAAGGC-3′
Gapdh_R: 5′-GGCATGGACTGTGGTCATGA-3′
HSF5_RTset1_F: 5′-AGGGCTACCATTCAGCACAC-3′
HSF5_RTset1_R: 5′-GACTTGTTAGCAGGCCCCAT-3′
Hspa1a_RTq_1F: 5′-CATCAGTGGGCTGTACCAGG-3′
Hspa1a_RTq_1R: 5′-AATGACAGTCCTCAAGGCCAC-3′
Hspa1b_RTq_1F: 5′-CCGACAAGGAGGAGTTCGTG-3′
Hspa1b_RTq_1R: 5′-CTTGACAGTAATCGGTGCCC-3′
Primer sequences are listed in Supplementary Data 1.

## Electrophoretic mobility shift assay (EMSA)

HSF5 N-terminal protein (aa1-209) was purified by Ni-NTA under native condition (20 mM Tris-HCl pH 7.5, 150 mM NaCl, 300 mM imidazole, 1% TritonX100). Purified protein was subjected to HiTrap Q column using AKTA pure system, and then the flowthrough fraction was collected. The flowthrough fraction was subjected to the HiTrap S column and eluted by a salt gradient from 0 to 1 M NaCl in a buffer (20 mM Tris-HCl pH 8.0, 1 mM 2-Mercaptoethanol). The eluted protein was dialyzed against a buffer (20 mM Tris-HCl pH 8.0, 100 mM NaCl, 1 mM 2-Mercaptoethanol, 10% glycerol).

The annealed synthetic oligonucleotide DNA was labeled with [α-$^{32}$P]dCTP (Perkin Elmer NEG013H250UC, 3000 Ci/mmol) by Klenow polymerase. $^{32}$P-labeled DNA was separated by electrophoresis in a 10% polyacrylamide gel, eluted from gel slices with an elution buffer containing 1 mM EDTA, and 10 mM Tris-HCl (pH 8.0), and precipitated by ethanol. The synthetic oligonucleotide DNA sequences are as follows; For the target (T) sequence probe,

HSF5-EMSA-T-up: 5′- AGGGCATAGATATGGAACTCTCTTAGAACT CTCTTAGAACTTACTACTG-3′

HSF5-EMSA-T-bottom: 5′- AGGCAGTAGTAAGTTCTAAGAGAGTTC TAAGAGAGTTCCATATCTATGC-3′

For the mutant (M) sequence probe,

HSF5-EMSA-M-up: 5′- AGGGCATAGATAACCTTGAGTCTATCTTGA GTCTATCTTGAAACTACTG-3′.

HSF5-EMSA-M- bottom: 5′- AGGCAGTAGTTTCAAGATAGACTCAA GATAGACTCAAGGTTATCTATGC-3′

The $^{32}$P-labeled DNA (0.04 pmol, $2 \times 10^4$ to $4 \times 10^4$ cpm) was mixed with 18.75-300 ng of HSF5 DNA binding domain in 10 µl of a buffer containing 20 mM Tris-HCl pH 8.0, 100 mM NaCl and 2 mM MgCl$_2$. The mixture was incubated at 25 °C for 10 min and loaded with glycerol dye mix (25% glycerol, 1 mM EDTA, 0.01% bromophenol blue) on an 8% polyacrylamide gel (acrylamide bis-acrylamide, 29:1) containing 89 mM Tris-borate (pH 8.3)−2 mM EDTA. After electrophoresis, the gel was dried and subjected to autoradiography by the Typhoon FLA7000 Biomolecular imager (Cytiva).

## SEC−MALS Experiments

The full length HSF5 expression construct was cloned into a pMAL-c6T vector (New England Biolabs) and fused to His$^6$-MBP and a tobacco etch virus protease cleavage site at the N-terminus. His-MBP-full length HSF5 was purified by HisTrap HP (Cytiva), and gel filtration using a Superdex 200 pg 16/600 column (Cytiva). Size exclusion chromatography with multi-angle light scattering (SEC−MALS) was performed using a DAWN HELEOS8+ (Wyatt Technology Corporation, Santa Barbara, CA, USA), a high-performance liquid chromatography pump LC-20AD (Shimadzu, Kyoto, Japan), a refractive index detector RID-20A (Shimadzu), and a UV−vis detector SPD-20A (Shimadzu), which were located downstream of the Shimadzu liquid chromatography system connected to a PROTEIN KW-803 gel filtration column (Cat. no. F6989103; Shodex, Tokyo, Japan). Differential RI (Shimadzu) downstream of MALS was used to determine the protein concentrations. The running buffer used contained 25 mM HEPES/KOH (pH 7.2) and

150 mM KCl. Approximately 100 μL of the sample was injected at a flow rate of 1.0 mL /min. Data was then analyzed using ASTRA version 7.0.1 (Wyatt Technology Corporation). Molar mass analysis was also performed over half of the width of the UV peak top height. 30 min after incubation at 42 °C, 100 μL of MBP-full length HSF5 (12.6 μM) or HSF5-DBD (26.6 μM) was injected.

## Preparation of testis extracts and immunoprecipitation

Testis chromatin-bound and -unbound extracts were prepared as described previously[69]. Briefly, testes were removed from male C57BL/6 mice (P18-23, 4 weeks and 8 weeks old), detunicated, and then resuspended in low salt extraction buffer (20 mM Tris-HCl pH 7.5, 100 mM KCl, 0.4 mM EDTA, 0.1% TritonX100, 10% glycerol, 1 mM β-mercaptoethanol) supplemented with Complete Protease Inhibitor (Roche). After homogenization, the soluble chromatin-unbound fraction was separated after centrifugation at 100,000 g for 10 min at 4 °C. The chromatin bound fraction was extracted from the insoluble pellet by high salt extraction buffer (20 mM HEPES-KOH pH 7.0, 400 mM KCl, 5 mM MgCl₂, 0.1% Tween20, 10% glycerol, 1 mM β-mercaptoethanol) supplemented with Complete Protease Inhibitor. The solubilized chromatin fraction was collected after centrifugation at 100,000 g for 10 min at 4 °C.

For immunoprecipitation of endogenous HSF5 from extracts, 5 μg of affinity-purified rabbit anti-HSF5-C, HSF5-N1, HSF5-N2 and control IgG antibodies were crosslinked to 50 μl of protein A-Dynabeads (Thermo-Fisher) by DMP (Sigma). The antibody-crosslinked beads were added to the testis extracts prepared above. The beads were washed with low salt extraction buffer. The bead-bound proteins were eluted with 40 μl of elution buffer (100 mM Glycine-HCl pH 2.5, 150 mM NaCl), and then neutralized with 4 μl of 1 M Tris-HCl pH 8.0. The immunoprecipitated proteins were run on 4-12% NuPAGE (Thermo-Fisher) in MOPS-SDS buffer and immunoblotted. Immunoblot images were developed using ECL prime (GE healthcare) and captured by FUSION Solo (VILBER).

## Mass spectrometry

The immunoprecipitated proteins were run on 4-12% NuPAGE (Thermo Fisher) by 1 cm from the well and stained with SimplyBlue (Thermo Fisher) for in-gel digestion. The gel containing proteins was excised, cut into approximately 1 mm sized pieces. Proteins in the gel pieces were reduced with DTT (Thermo Fisher), alkylated with iodoacetamide (Thermo Fisher), and digested with trypsin and Lysyl endopeptidase (Promega) in a buffer containing 40 mM ammonium bicarbonate, pH 8.0, overnight at 37 °C. The resultant peptides were analyzed on an Advance UHPLC system (ABRME1ichrom Bioscience) connected to a Q Exactive mass spectrometer (Thermo Fisher) processing the raw mass spectrum using Xcalibur (Thermo Fisher Scientific). The raw LC-MS/MS data was analyzed against the SwissProt protein/translated nucleotide database restricted to *Mus musculus* using Proteome Discoverer version 1.4 (Thermo Fisher) with the Mascot search engine version 2.5 (Matrix Science). A decoy database comprised of either randomized or reversed sequences in the target database was used for false discovery rate (FDR) estimation, and Percolator algorithm was used to evaluate false positives. Search results were filtered against 1% global FDR for high confidence level. All full lists of LC-MS/MS data are shown in Supplementary Data 7 (Excel file).

## ChIP-seq analysis

The seminiferous tubules from male C57BL/6 mice (P10-21) were minced and digested with accutase and 0.5 units/ml DNase II, followed by filtration through a 40 μm cell strainer (FALCON). Testicular cells were fixed in 1% formaldehyde (Thermo-Fisher) and 2 mM Disuccinimidyl glutarate (ProteoChem) in PBS for 10 min at room temperature. Crosslinked cells were lysed with LB1 (50 mM HEPES pH 7.5, 140 mM NaCl, 1 mM EDTA, 10% glycerol, 0.5% NP-40, and 0.25% Triton

X-100) and washed with LB2 (10 mM Tris-HCl pH 8.0, 200 mM NaCl, 1 mM EDTA, 0.5 mM EGTA). Chromatin lysates were prepared in LB3 (50 mM Tris-HCl pH8.0, 1% SDS, 10 mM EDTA, proteinase inhibitor cocktail (Sigma)), by sonication with Covaris S220 (Peak Incident Power, 175; Acoustic Duty Factor, 10%; Cycle Per Burst, 200; Treatment time, 600 sec; Cycle, 4).

HSF5 ChIP was performed using chromatin lysates and protein A Dyna-beads (Thermo-Fisher) coupled with 5 μg of rabbit anti-HSF5-C antibody and rabbit control IgG as described previously[11]. After 4 h of incubation at 4 °C, beads were washed 4 times in a low salt buffer (20 mM Tris-HCl (pH 8.0), 0.1% SDS, 1% (w/v) TritonX-100, 2 mM EDTA, 150 mM NaCl), and two times with a high salt buffer (20 mM Tris-HCl (pH 8.0), 0.1% SDS, 1% (w/v) TritonX-100, 2 mM EDTA, 500 mM NaCl). Chromatin complexes were eluted from the beads by agitation in elution buffer (10 mM Tris-HCl (pH 8.0), 300 mM NaCl, 5 mM EDTA, 1% SDS) and incubated overnight at 65 °C for reverse-crosslinking. Eluates were treated with RNase A and Proteinase K, and DNA was ethanol precipitated.

ChIP-seq libraries were prepared using 20 ng of input DNA, and 4 ng of ChIP DNA with KAPA Library Preparation Kit (KAPA Biosystems) and NimbleGen SeqCap Adaptor Kit A or B (Roche) and sequenced by Illumina Hiseq 1500 to obtain single end 50 nt reads.

ChIP-seq reads were trimmed to remove adapter sequence when converting to a fastq file. The trimmed ChIP-seq reads were mapped to the UCSC mm10 genome assemblies using Bowtie2 v2.3.4.1 with default parameters. Peak calling was performed using MACS program v2.1.0[70] (https://github.com/macs3-project/MACS) with the option (-g mm -p 0.00001). To calculate the number of overlapping peaks between control IgG and HSF5-C ChIP, we used bedtools program (v2.27.1)[71] (https://bedtools.readthedocs.io/en/latest/). ChIP binding regions were annotated with BED file using Cis-regulatory Element System (CEAS) v0.9.9.7 (package version 1.0.2), in which the gene annotation table was derived from UCSC mm10. Binding sites were classified using ChIPSeeker. Motif identification was performed using MEME-ChIP v5.1.1 website (http://meme-suite.org/tools/meme-chip)[72]. The motif database chosen was "JASPAR Vertebrates and UniPROBE Mouse". BigWig files, which indicate occupancy of HSF5, were generated using deepTools (v3.1.0) and visualized with Integrative Genomics Viewer software (v.2.8.3) http://software.broadinstitute.org/software/igv/home. Aggregation plots and heatmaps for each sample against the dataset of HSF5 target genes were made using deeptools program v3.5.0. Sequencing data are available at DDBJ Sequence Read Archive under the accession DRA017059.

## Cleavage under targets and tagmentation (CUT & Tag)

The single cell suspensions were prepared by incubating the minced seminiferous tubules from male C57BL/6 mice at P18-19 with collagenase for 10 min and Accuase (Innovative Cell Technologies, Inc.) for 5 min at room temperature. The 4c DNA content spermatocytes were collected from prepared single cell suspensions using DyeCycle Violet Dye and FACS method[44]. CUT&Tag was conducted according to the previous report with some modification[73,74]. Protein A-Tn5 fusion protein was purified through Chitin resin (NEB S6651S) from *E. coli* strain BL21-CodonPlus (DE3)-RIPL (Agilent) that was transformed with expression vector 3XFlag-pA-Tn5-Fl (Addgene plasmid #124601). Briefly, spermatocytes were fixed in 0.1% PFA in PBS for 2 min at room temperature. The reaction was stopped by adding 10% volumes of 2.5 M Glycine.

ConA-coated magnetic beads were prepared by incubating the Dynabeads MyOne Streptavidin T1 (invitrogen 65601) with biotin-conjugated concanavalin A (0.77 mg/ml, Sigma Aldrich, C2272) for 30 min at room temperature. The fixed cells were incubated with ConA magnetic beads for 30 min. at room temperature. ConA beads bound cells were incubated at 4 °C with primary antibodies overnight. After washing with Dig-wash buffer (0.05% Digitonin, 20 mM HEPES,

150 mM NaCl, 0.5 mM Spermidine with Complete Protease Inhibitor (Roche)), cells were incubated with secondary antibodies (guinea pig anti-rabbit IgG) at room temperature for 1 h. After three washes with Dig-wash buffer, cells were incubated with proteinA-Tn5 at room temperature for 1 h. Cells were incubated in Dig-wash buffer with 10 mM $MgCl_2$ at 37 °C for 1 h. The cells were incubated with 16.5 mM EDTA and 0.1% SDS at 65 °C overnight. After incubating with Protease K at 37 °C for 30 min, DNA was extracted with phenol-chloroform extraction and precipitated with ethanol. DNA was amplified by PCR using NEBNext High-Fidelity 2x PCR Master mix (New England Bio-Labs). The PCR cycle was as follows; 72 °C for 5 min, 98 °C for 30 sec, 98 °C for 10 sec, 63 °C for 20 sec and 72 °C for 1 min 98 °C for 10 sec. and 63 °C for 20 sec were repeated for 20 times for DNA tagmented with control IgG and 18 times for tagmented with HSF5-N1, HSF5-N2 and HSF5-C antibodies. Amplified DNAs were purified using Ampure XP beads. Purified DNAs were sequenced by Illumina NextSeq500 to obtain single-end 75nt reads.

## CUT&Tag data analysis

CUT&Tag data processing were conducted according to developer's instruction available online (Zheng et al., CUT&Tag Data processing and Analysis Tutorial. *Protocol.io* (2020) https://doi.org/10.17504/protocols.io.bjk2kkye). Sequenced reads were aligned to the mouse genome (mm10) and E. coli genome (BL21 (DE3)) using bowtie2 ver. 2.5.1 after trimming adapter sequence and to removing low-quality ends using Trim Galore! ver. 0.6.4._dev. The spikeIn calibration were carried based on the reads which mapped on E. coli genome. The peak calling was performed using control IgG data as control using SEACR ver. 1.3. Using bedtools, common peaks among HSF5-N1, N2 and C antibodies but not control IgG were obtained as HSF5-bound site. Motif identification was performed using MEME v5.5.4 website (http://meme-suite.org/tools/meme-chip) with following settings; -dna -oc. -nostatus -time 14400 -mod anr -nmotifs 10 -minw 6 -maxw 15 -objfun classic -revcomp -markov_order 0.

## SMART RNA-seq Analysis

Spermatocytes at zygotene/pachytene from WT and *Hsf5* KO male testis were sorted using SH800S cell sorter (SONY) using DyeCycle Violet (DCV) stain (Thermo Fisher)[44]. Total RNAs were prepared by Trizol (Thermo Fisher) and quality of total RNA was confirmed by BioAnalyzer 2100 (RIn > 8) (Agilent). cDNA was synthesized by SMART-Seq mRNA kit (Takara, 634772). Library DNAs were prepared according to the Nextera XT DNA Library Preparation Kit (Illumina, FC-131-1096) and sequenced by Illumina NextSeq 500 (Illumina) using Nextseq 500/550 High Output v2.5 Kit (Illumina) to obtain single end 75 nt reads.

The resulting reads were aligned to the mouse genome UCSC mm10 using STAR ver.2.7.8a after trimmed to remove adapter sequence and low-quality ends using Trim Galore! v0.6.6 (cutadapt v2.5). Differential expression analysis using TPM was done by RSEM v1.3.3, and DESeq2 (v1.36.0). GTF file was derived from UCSC mm10. Principal component analysis and GO-term analysis were performed using DAVID Bioinformatics Resources 6.8[75]. Sequencing data are available at DDBJ Sequence Read Archive under the accession DRA017058.

## ATAC-seq

Tn5-Mxe GyrA -intein-Chitin-Binding Domain fusion protein was purified through Chitin resin (NEB S6651S) from *E. coli* strain BL21-CodonPlus (DE3)-RIPL (Agilent) that was transformed with the expression vector pTXB1-Tn5 (Addgene plasmid #60240). Tn5 transposome assembly was done according to the previous paper[76].

The single cell suspensions were prepared by incubating the minced seminiferous tubules from male *Hsf5* +/- and *Hsf5* KO mice at P16 with collagenase for 10 min and Accuase (Innovative Cell

Technologies, Inc.) for 5 min at room temperature. The 4c DNA content spermatocytes were collected from prepared single cell suspensions using DyeCycle Violet Dye and FACS method[44].

ATAC-seq was conducted according to the previous report with some modification[77]. 100,000 cells were suspended in 50 μL of ice-cold lysis buffer with 10 mM Tris-HCl (pH 7.4), 10 mM NaCl, 3 mM MgCl2, 0.1% Tween-20, 0.1% NP-40 and 0.01% Digitonin. The cell suspension was incubated for 5 min on ice and centrifuged at 500 rpm for 5 min. Subsequently, 1 mL of ice-cold wash buffer (10 mM Tris-HCl [pH 7.4], 10 mM NaCl, 3 mM MgCl2, 0.1% Tween-20) was added. The suspension was incubated for 3 min on ice and centrifuged at 500 rpm for 10 min.

The pellet was resuspended in 50 μL of Transposition mix (25 μL 2 × Tagment DNA buffer; 20 mM Tris-HCl [pH7.6], 10 mM MgCl2, 20% Dimethyl Formamide, 1.25uL Tn5 transposase, 16.5ul PBS, 0.5ul 1% Digitonin, 0.5ul 10% Tween-20, 6.25ul H2O) (Cecchini et al.) and incubated at 37 °C for 30 min on a thermomixer with 1000 rpm mixing. The transposed DNA was purified with ChIP DNA Clean & Concentrator Kit (ZYMO RESEARCH, D5205). DNA samples were then amplified with PCR ([72 °C; 5 min] and [98 °C; 30 s] followed by 5 cycles of [98 °C; 10 s, 63 °C; 30 s, 72 °C; 1 min] using unique 8-bp dual indexes and NEBNext High-Fidelity 2 × PCR Master Kit (M0541S). Using 10% of the pre-amplified mixture, run qPCR to determine the number of additional cycles needed ([98 °C; 30 s] followed by 20 cycles of [98 °C; 10 s, 63 °C; 30 s, 72 °C; 1 min] using unique 8-bp dual indexes and NEBNext High-Fidelity 2 × PCR Master Kit (M0541S) and SYBR Green. After qPCR amplification, manually assess the amplification profiles. Using the remainder of the pre-amplification DNA, run 6 cycles (*Hsf5* +/-) and 5 cycles (*Hsf5* KO) determined required by qPCR.

Following the final amplification, DNA size selection was performed using solid-phase reversible immobilization (SPRI) beads (AMPure XP, Beckman Coulter, A63881) at an SPRI to DNA ratio of 0.6. The supernatant was further mixed with SPRI beads at a SPRI to DNA ratio of 1.0. The resulting supernatant was discarded, and the magnet-immobilized SPRI beads were washed with 80% ethanol. DNA was subsequently eluted in 20 μL of Tris-HCL (10 mM, pH8.0).

Purified DNAs were sequenced by Illumina NextSeq 500 using NextSeq 500/550 High Output v2 Kit (Illumina) to obtain single end 75 nt reads.

## Single-cell RNA-sequencing

Testes were collected from WT and *Hsf5* KO at P16. Testes were minced in 1.2 mg/ml type I collagenase (Wako) and incubated for 10 min at 37 °C, then tissue pellets were collected by centrifugation. Accutase (Innovative Cell Technologies, Inc.) was added to the tissue pellet and incubated for 5 min. at 37 °C. After incubation, DMEM with 10% FBS was added to block Accutase, and aggregate was disrupted by pipetting. Then, cell suspensions were filtered through a 95-μm and 35-μm cell strainer sieve (BD Bioscience). Cells were collected by centrifugation and re-suspended in PBS containing 0.1% BSA, and cells were collected using Cell Sorter SH800 (SONY) to remove dead cells. Collected cells were re-suspended in DMEM containing 10% FBS. Resulting approximately 10,000 single-cell suspensions were loaded on Chromium Controller (10X Genomics Inc.). Single cell RNA-seq libraries were generated using Chromium Single Cell 3' Reagent Kits v3 following manufacturer's instructions, and sequenced on an Illumina HiSeq X to acquire paired end 150 nt reads. The number of used embryos, the total numbers of single cells captured from testes, mean depth of reads per cell, average sequencing saturation (%), the number of detected genes, median UMI counts/cell, total number of cells before and after QC, are shown in Figure S5. Sequencing data are available at DDBJ Sequence Read Archive (DRA) under the accession DRA 017033.

## Statistical analysis of scRNA-seq

Fastq files were processed and aligned to the mouse mm10 transcriptome (GENCODE vM23/Ensembl 98) using the 10X Genomics Cell Ranger v 4.0 pipeline. Further analyzes were conducted on R (ver.3.6.2) via RStudio (ver.1.2.1335). Quality assessment of scRNA-seq data and primary analyzes were conducted using the Seurat package for R (v3.2.2)[78,79]. Only the cells that expressed more than 200 genes were used for further analysis to remove the effect of low-quality cells. The scRNA-seq data were merged and normalized using SCTransform function built in Seurat. The dimensional reduction analysis and visualization of cluster were conducted using RunUMAP function built in Seurat. The clustering of cells was conducted using FindNeighbors and FindClusters built in Seurat with default setting. Determination of DEGs was performed using FindAllMarkers function built in Seurat with following options, only.pos = T, min.pict = 0.25 and logfc.threshold = 0.25, for the identification of marker genes in each cluster, and FindMarker function built in Seurat with default settings to characterize the arbitrary group pf cells. To extract the germ cell population, the cell populations expressing germ cell marker, *Ddx4*, were selected. The selected cell populations were re-plotted on the UMAP. The cell populations showing the expression of somatic cell marker genes and the enrichment of mitochondrial genes were removed and the remaining cells were used as germ cell population. RNA velocity analysis was conducted using the RNA velocyto package for python (Python 3.7.3) and R with default settings[80], and visualized on UMAP plots built in Seurat.

## RNA-sequencing of heat shocked testes

The seminiferous tubules from whole testes of male *Hsf5* +/- (*n* = 3) and KO (*n* = 3) mice at P16 were unraveled and incubated in DMEM containing 10% FBS at 33 and 37 °C for 3 h. Total RNAs were prepared by RNeasy Mini Kit (QUIAGEN, 74104). Quality of total RNA was confirmed by BioAnalyzer 2100 (RI*n* > 9). Library DNAs were prepared according to the Illumina TruSeq protocol using TruSeq. Standard mRNA LT Sample Prep Kit (Illumina) and sequenced by Illumina NextSeq 500 (Illumina) using Nextseq 500/550 High Output v2.5 Kit (Illumina) to obtain single end 75 nt reads.

## Gene enrichment analysis

Gene enrichment analyzes were performed using Metascape[81] with default settings and the results were visualized using R and GraphPad Prism9. To characterize the feature of each cluster, top100 representative genes in each cluster were used.

## Public ChIP-seq Data and RNA-seq data analysis

MEIOSIN ChIP-seq data described in our previous study[16] was analyzed for the *Hsf5* locus. MEIOSIN binding site was shown along with genomic loci from Ensembl on the genome browser IGV. 10xGenomics scRNA-seq data of of mouse adult testis was derived from GEO: GSE109033[36]. Reanalyses of scRNA-seq data were conducted using the Seurat package for R (v.3.1.3)[79] and pseudotime analyses were conducted using monocle package for R: R (ver. 3.6.2), RStudio (ver.1.2.1335), and monocle (ver. 2.14.0)[82] following developer's tutorial. The tissue expression atlas of mouse *Hsf* gene paralogs are adapted from Expression Atlas (https://www.ebi.ac.uk/gxa/home), and the expression levels are shown using Microsoft Excel.

## Quantification and statistical analysis

Statistical analyzes, and production of graphs and plots were done using GraphPad Prism9 or Microsoft Excel (version 16.48).

## Materials and resources availability

Mouse lines generated in this study have been deposited to Center for Animal Resources and Development (CARD). CARD ID3259 for *Hsf5* knockout mouse #21 (C57BL/6-*Hsf5^em1^*), CARD ID3268 for *Hsf5* knockout mouse #5 (C57BL/6-*Hsf5^em2^*), CARD ID3269 for *Hsf5* knockout mouse #9 (C57BL/6-*Hsf5^em3^*), CARD ID3432 for *Hsf5-3xFLAG-HA* knockin mouse#3 (C57BL/6-*Hsf5^em1(Hsf5-3xFLAG-HA)^*), CARD ID3433 for *Hsf5-3xFLAG-HA* knockin mouse#6 (C57BL/6-*Hsf5^em2(Hsf5-3xFLAG-HA)^*).

The antibodies are available upon request. There are restrictions to the availability of antibodies due to the lack of an external centralized repository for its distribution and our need to maintain the stock. We are glad to share antibodies with reasonable compensation by requestor for its processing and shipping. All unique/stable reagents generated in this study are available from the Lead Contact with a completed Materials Transfer Agreement. Further information and requests for resources and reagents should be directed to and will be fulfilled by the Lead Contact, Kei-ichiro Ishiguro (ishiguro@kumamoto-u.ac.jp).

## Reporting summary

Further information on research design is available in the Nature Portfolio Reporting Summary linked to this article.

## Data availability

All data supporting the conclusions are present in the paper and the supplementary materials. Raw sequence data generated in this study were publicly available as of the date of publication. Sequencing data have been deposited in DDBJ Sequence Read Archive (DRA) under the accession DRA017033 for scRNA-seq data of P16 WT and *Hsf5* knockout testes, DRA017058 (DRR502790, DRR502791) for the SMART RNA-seq data of the P16 sorted meiotic prophase spermatocytes, DRA017059 (DRR502792-DRR502794) for ChIP-seq data of WT testes, PRJDB16509 (DRR530443-DRR530448) for CUT&Tag data of WT, *Hsf5* +/- and *Hsf5* KO spermatocytes, PRJDB16509 (DRR530449-DRR530460) for RNA-seq of whole testes of WT and knockout mice at 33 and 37 °C, DRR540205, DRR540206 for ATAC-seq data of the control and *Hsf5* KO spermatocytes, respectively. Mass spectral data of HSF5 IP have been deposited in Japan ProteOme STandard Repository (jPOSTrepo) under the accession PXD051085 for ProteomeXchange and JPST003018 for jPOST [https://repository.jpostdb.org/entry/JPST003018]. MEIOSIN ChIP-seq data was derived from DDBJ DRA007778. The scRNA-seq data of mouse adult testis was derived from GEO: GSE109033. Reference genome for scRNA-seq, mm10 was obtained from 10x genomics website: [https://support.10xgenomics.com/single-cell-gene-expression/software/release-notes/build#mm10_3.0.0]. The image data generated in this study have been deposited in the Figshare under [https://doi.org/10.6084/m9.figshare.24160956]. Source data are provided with this paper.

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

## Acknowledgements

The authors thank Drs. Shosei Yoshida, Hitoshi Niwa for discussion, Yuki Horisawa-Takada for technical assistance, Akihiko Sakashita and Satoshi Namekawa for sharing their technical information of DCV FACS sorting, Mitsuhiro Endoh for kindly providing purified Tn5 and proteinA-Tn5 proteins, Marry Ann Handel for provision of H1t antibody. This study was supported by the program of the Research for Inter-University Research Network for High Depth Omics, IMEG, Kumamoto University. This work was supported in part by KAKENHI grant (#20K22638) to R.S.; KAKENHI grant (#23K05641) to H.A.; KAKENHI grants (#19H05743, #20H03265, #20K21504, #22K19315, #23H00379, #24H02050, #16H06276 AdAMS, #22H04922 AdAMS) to K.I.; Grant from AMED PRIME (21gm6310021h0003) to K.I.; Grants from The Mitsubishi Foundation; The Naito Foundation; Astellas Foundation for Research on Metabolic Disorders; The Uehara Memorial Foundation; Takeda Science Foundation to K.I.

## Author contributions

S.Y., supervised by T.O. and E.K., performed most of the experiments and wrote the draft of the manuscript. R.S. performed scRNA-seq analysis, EMSA, informatics of RNA-seq, CUT&Tag, and ChIP-seq data. S.K. and T.S. performed SEC-MALS analysis. K.K. and H.A. assisted cytological analysis. S.I. assisted plasmid construction. K.A. generated mutant mice and performed IVF. N.T. performed MS analysis. S.F. performed histological analysis. K.Y. and S.U. assisted scRNA-seq and RNA-seq. K.I. conducted the study and wrote the manuscript. The experimental design and interpretation of data were conducted by S.Y., R.S., and K.I.

## Competing interests

The authors declare no competing interests.
