## [Peer Review File · Nature Communications]

Atypical heat shock transcription factor HSF5 is critical for male meiotic prophase under non-stress conditionsREVIEWER COMMENTS

Reviewer #1 (Remarks to the Author):

NCOMMS-23-42503 Review

Yoshimura and colleagues have uncovered new aspects of the importance of heat shock transcription factor HSF5 during meiotic prophase of mouse spermatogenesis. The manuscript is well written and ensures to include the most recent data presented on the relevant topics, including that of Barutc et al. 2023 (Heliyon) and Saju et al. 2018 (Cell Reports). Despite these two publications, there were many gaps left following these reports. Here, Yoshimura and colleagues present a comprehensive report determining that HSF5 is: 1) essential for the progression of spermatogenesis beyond the pachytene stage under non-stress conditions. 2) HSF5 binds to promoters of genes associated with chromatin organization. 3) Determined the novel DNA motif that HSF5 binds to. These findings are important for the field as they advance our understanding of the controlled transition during pachynema, which is critical to ensure chromosome recombination and organization prior to the onset of chromosome segregation.

Major comments:

- More comments and reference to their own experiments are needed to address the claims made by Barutc et al. and whether they are substantiated by the author's findings or refuted by them. This is particularly important for MSCI and SWI/SNF chromatin-remodeling observations.
- Localization studies need to be summarized with a histological schematic as seen in other publications (i.e. the stages of the seminiferous tubule with summarized localization of HSF1).
- Immunofluorescence microscopy assessment of chromatin spreads/tubule squashes of HSF5 co-stained with key elements of meiotic prophase are required to bolster their characterization of HSF5 expression and localization. Post-meiotic prophase localization stages should also be assessed.
- Based on the immunohistological assessment HSF5 localizes to the sex body. The investigators report that a main cluster of cells (cluster 2) is affected in Hsf5 KOs. However, very limited assessment of MSCI or proteins that are important for that process.
- There is a lack of follow-up of protein expression analyses of SWI/SNF chromatin-remodeling factors.
- There is no reference to Li et al. J Genet Genomics (2022) <https://doi.org/10.1016/j.jgg.2022.03.005>, which assesses the ZFP541-KCTD19 complex in mouse spermatogenesis.
- The authors stated that “Presumably, the absence of HSF5 indirectly delays DSB repair processes at the pachytene stage as a secondary effect of chromatin disorganization and aberrant gene expression”. However, the investigators do not assess chromatin organization and they should have ATACseq (or similar) data presented.

Minor comments:

- Check for typographical errors – e.g., Line 92 the comma is misplaced.

Reviewer #2 (Remarks to the Author):

Here, the authors generate and explore the male reproductive phenotype of Hsf5 KO mice. There was a clear and compelling loss of spermatocytes during pachynema. However, this closely resembles a study published by another group (Bartuc et al., 2023). It seems that this study went a bit further to identify DNA binding sites/sequence and putative protein binding factors. However, the authors did not follow those results to clearly identify specific defects in gene expression, chromatin configuration, etc that contributed to the pachynema arrest. As such, this reviewer feels this study does not significantly advance the current literature in terms of our understanding of the mechanistic role of HSF5 in meiosis.

Comments, largely in order of appearance, to assist the authors in a revision:

This reviewer appreciates that English is not the authors' first language. Overall, the manuscript is written fairly well. However, there are multiple places where grammar should be improved.

Abstract

25-26 So, is this gene expression change NOT occurring in females, as implied in the first sentence?

Would recommend present or past tense

36 Is "enforces" the correct term? This was not clearly shown based on the data presented

Introduction

40 Occurs prior to the formation of sperm and oocytes, not sure it's fair to say "produces"

46 Replace "sperm production" with spermatid morphogenesis or spermiogenesis

62-3 Was Hsf5 identified in this or a previous study? Would clarify...

67-8 This sentence was confusing – "common in humans and mice" and then "humans and mice additionally"...

Can the relationship between HSFs and HSPs be more clearly communicated?

78 Sentences ending and starting on this line seem unrelated; the transition to a description of reproductive phenotypes needs some attention

85 "homolog" to ortholog

90-1 This is an unfair and rather misleading summary of a well-written and fairly conclusive study published 6 months ago – those authors reported the expression pattern, male reproductive phenotype, and mechanism of germ cell loss in those mice, which were also infertile

92-4 Sentences need to be smoothed out

99-101 ScRNA-seq probably not the most conclusive way to highlight a block in pachynema

Results

113 Replace "directs" with a weaker word/phrase – this is too strong based on that report; perhaps 'is required during'

114 Replace "suggested" with a stronger word – didn't the ChIP-seq data convincingly show binding?

116 Same comment, why use "suggested"? Would say 'revealed'

118 To be consistent with verbiage in your previous report on MEIOSIN, meiotic entry occurs during the first wave at P8, in preleptonema

121-38 “Suggesting” again used many times – this makes this reviewer curious, how sure are the authors about the deductions from these results? Can this word be replaced with others that may be more definitive? Do the data support more definitive statements?

141-163 throughout this section, would replace “expression patterns” and “expression profiles” with ‘steady-state mRNA abundance/levels’ – that’s all RT-PCR is capable of measuring, and semi-quantitatively at that. Gene expression programs include transcription, splicing, RNA stability, translation efficiency, protein stability, etc. Here, it appears all that is being assessed/discussed is steady-state mRNA levels. Which can correlate to protein levels, albeit only about half the time based on genome-wide comparisons in cell lines (e.g., PMID: 21593866); male germ cells have numerous examples of post-transcriptional control, so the correlation between mRNA and protein levels may be even lower than in somatic cells

160 replace “embryonic” with ‘fetal’; and how can the authors discount “barely detected”? Perhaps this is because there are so many fewer oocytes than spermatocytes analyzed? Or perhaps the low levels of mRNAs in the ovary are meaningful? Would re-phrase

162-3 Don’t need “suggest” and “may” in the same sentence

111-163 Can this be condensed? There was a lot of verbiage covering minimal results

168-9 vs 118 STRA8 as “a marker of meiotic initiation” – it is expressed in preleptonema – again, to tie back to comment on 118, should be consistent with meiotic initiation/entry – is it during preleptonema or leptonema?

170-1 This didn’t make much sense as written – “persisted in the round spermatids of stage VI tubules” ... was it not present in other round spermatids? If not, how did it “persist”?

176-7 Does immunostaining first wave spermatocytes “confirm” results from steady-state? Or just complement?

177-8 mRNA levels increased, could be due to increased transcription or increased stability

180 so, if potentially post-transcriptionally regulated, how meaningful are the results from the lengthy text in lines 111-163?

202 “severely abolished” = redundant

203 “seemingly”?

205 Again, delete “suggest” – the results PROVE, CONCLUDE, REVEAL

111-369 At this point in the manuscript, it is not clear if or how these results advance those from Bartuc et al., published in April 2023.

371 “the genes” – please specify, which ones?

374 From the text, it is not clear if these 165 sites are the only ones in the entire genome, or just the ones with promoter proximity – revise text to clarify

405-6 curious what is meant specifically by “stem cell population maintenance” means as a category – for example, housekeeping genes would fit into this category... e.g., actin/tubulin/laminin are each required for maintenance of stem cell populations. The authors alluded to this in 406-7.

416-28 mRNA levels, not “expression patterns”, and is it fair to say any of these are “derepressed”? Based on what?

427 this conclusion is rather lackluster as written – “suggest... positively or negatively regulates”

416-28 Perhaps this reviewer is misinterpreting the text, but since the authors conclude HSF5 protein is only detectable in spermatocytes, then how are “HSF5-bound genes were higher in spermatogonia”? Perhaps this just needs to be rewritten, but these genes should not be bound by HSF5 in spermatogonia bc it’s not expressed in those cells

What specific changes in the mRNA abundance of HSF5-bound genes occurs in pachynema? This seems like a critical missing component here – what is the outcome in terms of target genes in the absence of HSF5?

What are the changes in chromatin during pachynema in Hsf5 KO spermatocytes?

430-445 While quite interesting, these results are rather preliminary and are perhaps not the strongest to finish the results section with... what do they mean in terms of chromatin organization in pachynema? Can the authors show concomitant SWI/SNF chromatin-expected changes in pachynema?

Discussion

450-462 It is typical for this first paragraph to be a summation of the results of this study and their importance

The authors should thoroughly highlight the advancements in this study in comparison to those from Bartuc et al., published in April 2023. As written, the authors minimize this previous study to the point where it almost seems like those authors didn't already present ~half of what is shown here

Reviewer #3 (Remarks to the Author):

This manuscript entitled “Atypical heat shock transcription factor HSF5 is critical for male meiotic prophase under non-stress conditions” by Yoshimura and colleagues investigate the mechanism underlying how HSF5 depletion in germ cells causes defects in spermatogenesis. The authors found the expression of this testis-specific heat shock protein HSF5 increased abruptly from the mid-pachytene spermatocytes, which coordinates its essential function in driving developmental progression beyond the mid-pachytene stage during spermatogenesis. The authors employed a range of methodologies including SMART RNA-seq, single-cell RNA-seq (scRNA-seq) and CHIP-seq to investigate the potential impact of HSF5 depletion on gene expression, and whether the genome-wide binding of HSF5 correlates with differentially expressed genes upon HSF5 knockout.

Many germ cell-specific proteins including a subset of transcription factors have been published about their requirement for meiotic progression beyond the pachytene stage, other HSF members have been reported to play a role in spermatogenesis under stress or non-stress conditions, and HSF5 has been shown to be essential for meiosis and male infertility in zebrafish (Saju et al. 2018) and mice (Barutc et

al. 2023). Nevertheless, this study provides more mechanistic insights into the action of HSF5, and these interesting findings have the potential to constitute a valuable addition to the current knowledge.

This manuscript is overall well designed and well written. While many results have been convincing, the following concerns need to be addressed before further consideration of acceptance:

Major concerns

1. Lines 188-191: Since HSF5 protein is absent in KO testes, it needs to be explained why immunofluorescent signals are present in some KO seminiferous tubules and even not diminished compared to those in WT (Fig. 2C). If the antibody-raised signals are somehow not completely specific to HSF5 at all stages, the immunostaining results reflecting the stage-dependent cellular distribution of HSF5 should be reevaluated (Fig. 1F).
2. Lines 243-246: The spermatocytes in the selected images look like to be at late zygotene stage (Fig. 3E) - why not showing pachytene spermatocytes with an aberrant number of DMC1 foci in KO? The statistic result is inconsistent with about 50 DMC1 foci observed for each pachytene cell in WT (Luo et al., Nat Commun 2013).
3. Lines 355-364, Fig. 5: It remains unknown about the expression of autosomal and sex chromosome-linked genes in Cluster 10 population of Hsf5 KO, since this subtype was largely absent in Hsf5 KO testis. Although Cluster 2 population exhibited downregulation of sex chromosome-linked genes, the defects in meiotic silencing of the sex chromosomes might exist in Hsf5 KO testis. Have the authors checked the distribution of MSCI factors in H1T-positive mid-late pachytene spermatocytes using spread analysis?
4. One intriguing finding of this manuscript is that scRNA-seq analyses reveal the expression levels of autosomal genes are elevated in Hsf5 KO Cluster 2 subpopulation. Have the authors compared the average expression level of autosomal and sex chromosome-linked genes in Hsf5 KO spermatocytes with wild-type cells from SMART RNA-seq datasets.
5. Lines 364-367: As for the claim "...KO spermatocytes exhibited gene expression patterns that were comparable to those in WT...", how about the gene expression differences seen as well from other Clusters (see those statistically meaningful based on the statistic labels), including sex chromosomal genes in Clusters 0 and 1 (Fig. 5I) and autosomal genes in Clusters 3 and 4 (Fig. 5J)? It's stunning that the spermatocyte loss in Clusters 2 and 10 was due to gene expression differences at the same stage (Cluster 2) when HSF5 protein undergoes an expression burst rather than being due to those at earlier stages (prior to Cluster 2). Suggest to analyze deeply the gene expression profiles of earlier Clusters, especially Cluster 3, when cell populations were not yet sharply diminished.
6. It can be interesting to cross-analyze the SMART RNA-seq data and scRNA-seq data, as it's reasonable to check all the 958 downregulated genes in Hsf5 KO (SMART RNA-seq) through several related Clusters (scRNA-seq). How many genes are mutually inclusive between two datasets?
7. Lines 372-377, Fig. 6A and B: Were total testis samples collected for ChIP-seq? HSF5 binding targets seems quite limited in testis compared with other typical transcription factors, what parameters are used

for HSF5 genome-wide peak calling? These information should be added in the text or figure legend.

8. Lines 406-414: How could Hsf5 KO spermatocytes be rapidly eliminated at the pachytene stage, undergoing cell apoptosis or arrest to affect cell populations? The remaining cells of any same subtype surviving such elimination may be sufficient for transcriptomic analyses.

9. Lines 416-428: Are any individual HSF5-bound genes (Classes C1 to C7), whether they are expressed at relatively high or low levels in any certain-stage germ cell population (Clusters 0 to 11) (Fig. 6K), dysregulated in Hsf5 KO at a given stage?

10. Lines 430-436: The immunoprecipitation and mass spectrometry analysis with three different HSF5 antibodies identified that SMARCA5 and SMARCC2 proteins interacted with HSF5. This result needs to be confirmed by Co-IP using in vivo or in vitro system.

Minor concerns

1. Line 59: The citation (Kojima et al. 2019) is missing in the reference list.

2. Lines 119-120: As introduced by the authors, the biological function of HSF5 has already been reported elsewhere.

3. Fig. 1F: Some images look weird, for instance, the inner cells stained as red (for SYCP3) should be non-specific signals at stages IV-V and VI.

4. Fig. 3F: Are the labels reversed between SYCP3 and MLH1? Red foci should be MLH1 and not clearly seen on the images.

5. Lines 322-330: What were the germ cell subtypes designated for Clusters 4, 5, and 6?

6. Lines 339-341: In Cluster 2, spermatocytes seem NOT totally absent in KO (Fig. 5C).

7. Lines 355-358: In Cluster 10, there seems very few spermatocytes that can be analyzed in KO (Fig. 5C), thus “abrupt downregulation” may be not properly claimed. Then, is it logical to suggest that “those Cluster 2 represents...”?

8. Lines 367-369: Suggest to rewrite this conclusion towards the angle of gene expression regulation by HSF5.

9. Discussion: Suggest to reduce redundant comments, for example, lines 496-498.

NCOMMS-23-42503 REVIEWER COMMENTS

Reviewer #1

Yoshimura and colleagues have uncovered new aspects of the importance of heat shock transcription factor HSF5 during meiotic prophase of mouse spermatogenesis. The manuscript is well written and ensures to include the most recent data presented on the relevant topics, including that of Barutc et al. 2023 (Heliyon) and Saju et al. 2018 (Cell Reports). Despite these two publications, there were many gaps left following these reports. Here, Yoshimura and colleagues present a comprehensive report determining that HSF5 is: 1) essential for the progression of spermatogenesis beyond the pachytene stage under non-stress conditions. 2) HSF5 binds to promoters of genes associated with chromatin organization. 3) Determined the novel DNA motif that HSF5 binds to. These findings are important for the field as they advance our understanding of the controlled transition during pachynema, which is critical to ensure chromosome recombination and organization prior to the onset of chromosome segregation.

Major comments:

- More comments and reference to their own experiments are needed to address the claims made by Barutc et al. and whether they are substantiated by the author's findings or refuted by them. This is particularly important for MSCI and SWI/SNF chromatin-remodeling observations.

While we partially agree with the findings of Barutc et al. (2023), our study reveals several differences in conclusions and interpretations as outlined below.

(1) Barutc et al. observed HSF5 appearing at the pachytene stage, mostly localizing to the XY body in spermatocytes. In contrast, our study shows that HSF5 localizes over the nuclei in pachytene spermatocytes and to the XY body, particularly in stage VII-VIII pachytene spermatocytes (Fig. 2a, 2d, Fig. S2a).

(2) In the Figure 3C (Barutc et al), despite that appreciable number of spermatocytes with normal XY body exist, there was no description about it and no statistical data were presented. Barutc et al. concluded that Hsf5 knockout (KO) pachytene spermatocytes exhibited elongated XY chromosomes rather than forming a condensed XY body. Our study, illustrated in Figure 4b, indicates that most Hsf5 KO pachytene spermatocytes exhibit a typical XY body, and the remaining ~24.2% show dispersed γ H2AX signals (Fig. 4d). Contrary to Barutc et al., our SMART-seq data (Fig.S4) and scRNA-seq data (Fig. 6j) do not show upregulation of sex chromosomal genes in Hsf5 KO pachytene spermatocytes. Our interpretation is that Hsf5 KO pachytene spermatocytes form the XY body and establish meiotic sex chromosome inactivation (MSCI), as pol II was excluded from the XY body in Hsf5 KO pachytene spermatocytes (Fig. 4c).

(3) In Barut et al., SMARCA4, a subunit of the SWI/SNF chromatin-remodeling complex, was reported as a downregulated gene in Hsf5 KO pachytene spermatocytes. However, our SMART-seq data do not show downregulation of *Smarca4* or any other genes encoding SWI/SNF components. The observed discrepancies could be attributed to different methodologies for isolating spermatocytes: fluorescent sorting of DCV-stained spermatocytes from P17 testes in our study versus the isolation of spermatocytes from testes of unknown age by STA-PUT in Barut et al. (2023), leading to potential differences in cellular populations between wild type (WT) and Hsf5 KO.

We have now included a discussion on these differing observations between the two studies (Line 523-559)."

- Localization studies need to be summarized with a histological schematic as seen in other publications (i.e. the stages of the seminiferous tubule with summarized localization of HSF1).

We added a histological schematic that summarizes localization of HSF5 and H1t together with HSF1 that was published (Åkerfelt et al, 2010b) in the stages of the seminiferous tubules in the new Figure 2f.

- Immunofluorescence microscopy assessment of chromatin spreads/tubule squashes of HSF5 co-stained with key elements of meiotic prophase are required to bolster their characterization of HSF5 expression and localization. Post-meiotic prophase localization stages should also be assessed.

We added new data in the Fig. S2b, S2c, S2d, S2e.

Our close inspection of HSF5 co-stained with γ H2AX using tubule squashes revealed even distribution of the HSF5 signals over nuclei in most of pachytene spermatocytes (Fig. S2b, S2c).

Further, we showed that an intense signal of HSF5 which coincided with γ H2AX in some, if not all, pachytene spermatocytes particularly at stage VII-VIII. This analysis was done using pachytene spermatocytes squashed from the stage VII-VIII seminiferous tubule excised from *Stra8-3xFLAG-HA-p2A-GFP* knock-in mouse (Fig. S2d, S2e).

Post-meiotic prophase localization stages were assessed in Fig2a and summarized in Fig2f.

- Based on the immunohistological assessment HSF5 localizes to the sex body. The investigators report that a main cluster of cells (cluster 2) is affected in Hsf5 KOs. However, very limited assessment of MSCI or proteins that are important for that process. The HSF5 signal was observed over the nuclei of pachytene spermatocytes in the stage VI-XII tubules (Fig. 2a, 2d). We noticed that an intense signal of HSF5 appeared in pachytene spermatocytes particularly at stage VII-VIII, which coincided with γ H2AX signal. Now, we newly showed that the same immunostaining patterns of HSF5 signal were confirmed by other different HSF5 antibodies (HSF5-N1, HSF5-N2) (Fig. S2a).

Further, the same pattern was verified by immunostaining using HA antibody in the seminiferous tubules of *Hsf5-3xFLAG-HA* knock-in mouse (Fig. 2d).

However, as mentioned above, (1) our close inspection of HSF5 co-stained with γ H2AX using isolated pachytene spermatocytes revealed even distribution of the HSF5 signals over nuclei in most pachytene spermatocytes, and not all the pachytene spermatocytes showed HSF5 association to the XY body (Fig. S2b, S2c, S2d, S2e). (2) We did not see upregulation of sex chromosomal genes in our SMART-seq data (Fig. S4) nor scRNA-seq data in *Hsf5* KO pachytene spermatocytes (Fig 6j). (3) We observed that ~75.8% of *Hsf5* KO pachytene spermatocytes exhibited a typical γ H2AX signal on the XY body (Fig. 4b, Fig. S2b). (4) Pol II was excluded from XY body in *Hsf5* KO pachytene spermatocytes (Fig 4c). Thus, our interpretation is that *Hsf5* KO pachytene spermatocytes formed the XY body and establish MSCI. We added our interpretation in the text (Line 244-251) and in the discussion (Line 523-559)

- There is a lack of follow-up of protein expression analyses of SWI/SNF chromatin-remodeling factors.

We tried to validate the possible interaction of HSF5-SWI/SNF chromatin-remodeling factors by HSF5 IP-WB analysis, and found that SMARCA5, a SWI/SNF subunit, was under detection level by WB. One interpretation is that HSF5 interacts with the SWI/SNF complex weakly, consistent with lower detection levels of SWI/SNF subunits by MS spec analyses in the HSF5 immunoprecipitates. This is also the case for HSF5-KCTD19 interaction. Although these data were not strong enough to account for the potential role of HSF5 in the gene activation and repression, we thought the HSF5 IP-MS data are still informative for the field. Thus, we discussed the possibility that those interactions of HSF5 with SWI/SNF and KCTD19 could be transient or regulatory rather than stoichiometric in the discussion part (Line 500-505, Line 512-518).

- There is no reference to Li et al. J Genet Genomics (2022) <https://doi.org/10.1016/j.jgg.2022.03.005>, which assesses the ZFP541-KCTD19 complex in mouse spermatogenesis.

We added the reference Li et al. J Genet Genomics (2022).

- The authors stated that “Presumably, the absence of HSF5 indirectly delays DSB repair processes at the pachytene stage as a secondary effect of chromatin disorganization and aberrant gene expression”. However, the investigators do not assess chromatin organization and they should have ATACseq (or similar) data presented.

We examined the changes in chromatin during pachynema in *Hsf5* KO spermatocytes by ATAC-seq. This analysis indicated that overall chromatin accessibility is slightly increased at TSSs in HSF5-target genes in *Hsf5* KO. The same trend was observed in class 1,2,3 of the HSF5-target genes.

Thus, although we may discuss subtle changes in chromatin in *Hsf5* KO, we assume that assessing open chromatin in *Hsf5* KO versus control spermatocytes would add little new information to our paper because of limited interpretation the data. This data is solely to show the reviewers.

Now we removed the relevant sentence “as a secondary effect of chromatin disorganization” (Line658).

Minor comments:

- Check for typographical errors – e.g., Line 92 the comma is misplaced.

We corrected this.

Reviewer #2 (Remarks to the Author):

Here, the authors generate and explore the male reproductive phenotype of *Hsf5* KO mice. There was a clear and compelling loss of spermatocytes during pachynema. However, this closely resembles a study published by another group (Bartuc et al., 2023). It seems that this study went a bit further to identify DNA binding sites/sequence and putative protein binding factors. However, the authors did not follow those results to clearly identify specific defects in gene expression, chromatin configuration, etc that contributed to the pachynema arrest. As such, this reviewer feels this study does not significantly advance the current literature in terms of our understanding of the mechanistic role of HSF5 in meiosis.

Comments, largely in order of appearance, to assist the authors in a revision:

This reviewer appreciates that English is not the authors' first language. Overall, the manuscript is written fairly well. However, there are multiple places where grammar should be improved.

Abstract

25-26 So, is this gene expression change NOT occurring in females, as implied in the first sentence?

Would recommend present or past tense

We corrected the sentences in the Abstract to present tense.

36 Is “enforces” the correct term? This was not clearly shown based on the data presented

We rephrased it as following “that is required for the gene expression program for pachytene transition during meiotic prophase in males”.

Introduction

40 Occurs prior to the formation of sperm and oocytes, not sure it's fair to say “produces”

We rephrased it as following “Meiosis occurs prior to the formation of sperm and oocytes.

46 Replace “sperm production” with spermatid morphogenesis or spermiogenesis

“sperm production” was replaced with spermatid morphogenesis.

62-3 Was Hsf5 identified in this or a previous study? Would clarify...

We rephrased as following “In the previous study (Ishiguro et al., 2020), we identified the Heat shock transcription factor family 5 (*Hsf5*) gene as one of the MEIOSIN/STRA8 target genes”.

67-8 This sentence was confusing – “common in humans and mice” and then “humans and mice additionally”...

We rephrased as following “HSF1, HSF2, HSF4, HSF5, and HSFY are common to human and mouse, with an additional HSFX for human and HSF3 for mouse”.

Can the relationship between HSFs and HSPs be more clearly communicated?

We rephrased as following “The best known HSF paralog associated with the stress response is HSF1, which is present in the cytoplasm as an inactive monomer when bound to HSPs. Upon sensing stress, HSF1 forms a homotrimer that trans-locates into the nucleus, binds specifically to the heat shock element (HSE) in the genome, and activates

HSR gene transcription". (Line 73-77)

78 Sentences ending and starting on this line seem unrelated; the transition to a description of reproductive phenotypes needs some attention

We rephrases it as following "Under non-stress conditions, the HSF family is known to regulate developmental processes of spermatogenesis"

85 "homolog" to ortholog
"homolog" was replaced with "ortholog".

90-1 This is an unfair and rather misleading summary of a well-written and fairly conclusive study published 6 months ago – those authors reported the expression pattern, male reproductive phenotype, and mechanism of germ cell loss in those mice, which were also infertile

Now we summarized the previous study as following : "it has been demonstrated that disruption of *Hsf5* led to apoptosis during spermatogenesis in mice with failure of MSCI and consequent infertility (Barutc et al. 2023)." (Line 92-93)

92-4 Sentences need to be smoothed out

We rephrased these sentences as following "it remained elusive which processes of meiosis HSF5 is involved in and whether HSF5 plays overlapping and/or different roles to other HSFs in meiotic prophase progression in the testis."

99-101 ScRNA-seq probably not the most conclusive way to highlight a block in pachynema

"single-cell (sc) RNA- seq analysis" was replaced with 'our genetic analysis of *Hsf5* KO'.

Results

113 Replace "directs" with a weaker word/phrase – this is too strong based on that report; perhaps 'is required during'

"directs" was now replaced with 'is required during'.

114 Replace "suggested" with a stronger word – didn't the ChIP-seq data convincingly show binding?

"suggested" was now replaced with 'demonstrated'.

116 Same comment, why use “suggested”? Would say ‘revealed’
“suggested” was now replaced by ‘revealed’.

118 To be consistent with verbiage in your previous report on MEIOSIN, meiotic entry occurs during the first wave at P8, in preleptonema

Now we removed this statement that was redundantly mentioned in the introduction.

121-38 “Suggesting” again used many times – this makes this reviewer curious, how sure are the authors about the deductions from these results? Can this word be replaced with others that may be more definitive? Do the data support more definitive statements?

Now we stated the conclusions in a definitive way and removed “suggesting” in these lines.

Accordingly, we revised as following : “the purified His-MBP-fused full length HSF5 protein was eluted at the peaks of approximate molar mass of 160 kDa and 340 kDa at 25 °C, and at the peak with megadalton-scale size after 42 °C treatment (Fig. 1C). This observation indicates that while HSF5 exists as a mixture of monomer and dimer/trimer at 25 °C, it forms high-order oligomers at higher temperature.” (Line 125-130)

141-163 throughout this section, would replace “expression patterns” and “expression profiles” with ‘steady-state mRNA abundance/levels’ – that’s all RT-PCR is capable of measuring, and semi-quantitatively at that. Gene expression programs include transcription, splicing, RNA stability, translation efficiency, protein stability, etc. Here, it appears all that is being assessed/discussed is steady-state mRNA levels. Which can correlate to protein levels, albeit only about half the time based on genome-wide comparisons in cell lines (e.g., PMID: 21593866); male germ cells have numerous examples of post-transcriptional control, so the correlation between mRNA and protein levels may be even lower than in somatic cells

We appreciated the reviewer’s suggestion. Now we replaced “expression patterns” and “expression profiles” with ‘steady-state mRNA abundance/levels. Accordingly, we revised the relevant sentences and words (Line 138-148).

160 replace “embryonic” with ‘fetal’; and how can the authors discount “barely detected”? Perhaps this is because there are so many fewer oocytes than spermatocytes analyzed? Or perhaps the low levels of mRNAs in the ovary are meaningful? Would rephrase

“embryonic” was replaced with ‘fetal’. Now we rephrased the sentence to “*Hsf5* expression was hardly detected in E12.5–E18.5 fetal ovaries by RT-qPCR (Fig.1e), suggesting *Hsf5* was expressed at a low level in fetal ovaries.”

162-3 Don’t need “suggest” and “may” in the same sentence
We removed “may”.

111-163 Can this be condensed? There was a lot of verbiage covering minimal results

Now we overall condensed this part, removing redundant statements.

168-9 vs 118 STRA8 as “a marker of meiotic initiation” – it is expressed in preleptonema – again, to tie back to comment on 118, should be consistent with meiotic initiation/entry – is it during preleptonema or leptonema?

We rephrased “a marker of meiotic initiation” as “a marker of preleptonema”.

170-1 This didn't make much sense as written – “persisted in the round spermatids of stage VI tubules”... was it not present in other round spermatids? If not, how did it “persist”?

We rephrased it as “The HSF5 signal began to appear in the mid-pachytene spermatocyte nuclei at stage VI. Then HSF5 was persistently observed in the spermatocyte nuclei of stage VII-XII seminiferous tubules, and in the round spermatids of stage I-VI”.

176-7 Does immunostaining first wave spermatocytes “confirm” results from steady-state? Or just complement?

We removed the sentence “confirming the observation above”.

177-8 mRNA levels increased, could be due to increased transcription or increased stability

We rephrased the sentence as “stability of *Hsf5* mRNA levels may be increased at the pachytene stage”.

180 so, if potentially post-transcriptionally regulated, how meaningful are the results from the lengthy text in lines 111-163?

As mentioned above, now we overall condensed statements (Line 111-163 in previous manuscript) and removed the statement.

202 “severely abolished” = redundant

We removed “severely”.

203 “seemingly”?

We removed “seemingly”.

205 Again, delete “suggest” – the results PROVE, CONCLUDE, REVEAL

We rephrased the sentence as “These results reveal that.....”

111-369 At this point in the manuscript, it is not clear if or how these results advance those from Bartuc et al., published in April 2023.

We conducted scRNA-seq analysis to identify the subtypes of *Hsf5* KO spermatocytes accompanied the alteration of gene expression profiles (Fig. 6), which was not addressed in the previous study (Barutic et al. 2023). Our scRNA-seq analysis advances bulk RNA-

seq done by Bartuc et al., because scRNA-seq analysis complements the limitation of bulk RNA-seq in which subtle differences in developmental cellular populations between control WT and *Hsf5* KO spermatocytes may have potentially contributed to bulk transcriptomic differences in gene expression.

Further, our cytological analyses show precise comparison of phenotypes in DSB repair processes (Fig. 4), which was not clearly shown in the previous report.

As described above, while partly agreeing with the previous finding (Barutc et al. 2023), our study show several differences to the previous study of *Hsf5* KO mice as described below.

(1) In the Barutc et al, HSF5 appears at the pachytene stage and mostly localizes to XY body in spermatocytes.

In our study, HSF5 localizes over the nuclei in pachytene spermatocytes, and to the XY body particularly in the stage VII-VIII pachytene spermatocytes (Fig. 2a, 2d, S2a, S2d, S2e).

(2) In the Barutc et al, they conclude that *Hsf5* KO pachytene spermatocytes exhibited elongated XY chromosomes rather than forming a condensed XY body. The previous study concluded that *Hsf5* KO pachytene spermatocytes showed a failure of MSCI and upregulation of sex chromosomal genes as revealed by RNA-seq data.

In the Figure 3C (Barutc et al), Despite that appreciable number of spermatocytes with normal XY body exist, there is no statistical data.

However, in our study, most of *Hsf5* KO pachytene spermatocytes exhibit a typical XY body (Fig 4b), where pol II was excluded (Fig 4c). We did not see upregulation of sex chromosomal genes in our SMART-seq data (Fig. S4) nor scRNA-seq data in *Hsf5* KO pachytene spermatocytes (Fig. 6j). Thus, our interpretation is that *Hsf5* KO pachytene spermatocytes form the XY body and establish MSCI.

(3) In the Barutc et al, SMARCA4, a subunit of SWI/SNF chromatin-remodeling complex, was downregulated in *Hsf5* KO pachytene spermatocytes. However, we did not see downregulation of *Sarca4* nor any of other SWI/SNF components in our SMART-seq data (Fig. S4). Although we do not know the exact reason for the different observations between previous study and ours, this could be due to different methodologies for isolation of spermatocytes : florescent sorting of DCV stained spermatocytes from P17 testes in our study versus isolation of spermatocytes from unknown age testes by STA-PUT (Barutc et al. 2023), which may have caused difference of cellular populations between WT and *Hsf5* KO.

Now we added discussion on these different observations between the two studies (Line 523-559).

371 “the genes” – please specify, which ones?

We changed the caption as follows; HSF5 alters gene expression pattern by binding to promoter regions with a unique target specificity

374 From the text, it is not clear if these 165 sites are the only ones in the entire genome, or just the ones with promoter proximity – revise text to clarify

We rephrase it as following “HSF5 bound to 165 sites across the genome”

405-6 curious what is meant specifically by “stem cell population maintenance” means as a category – for example, housekeeping genes would fit into this category... e.g., actin/tubulin/laminin are each required for maintenance of stem cell populations. The authors alluded to this in 406-7.

Now we added HSF5 CUT&Tag results in the new Fig. 7 and revised manuscript. Accordingly, this part was removed.

416-28 mRNA levels, not “expression patterns”, and is it fair to say any of these are “derepressed”? Based on what?

We replaced “expression patterns” with ‘alteration of mRNA levels’ (Line 425, 428).

As this reviewer pointed out, “derepressed” is not appropriate here. Therefore, we replaced “derepressed” with ‘upregulated’ (Line 401).

427 this conclusion is rather lackluster as written – “suggest... positively or negatively regulates”

We rephrased “HSF5 positively and negatively regulates the expression of its HSF5-target genes”. (Line 452-454)

416-28 Perhaps this reviewer is misinterpreting the text, but since the authors conclude HSF5 protein is only detectable in spermatocytes, then how are “HSF5-bound genes were higher in spermatogonia”? Perhaps this just needs to be rewritten, but these genes should not be bound by HSF5 in spermatogonia bc it’s not expressed in those cells

As we mentioned in the previous manuscript, HSF5 both positively and negatively regulate its target genes. Fig.61 (previous manuscript) suggests that HSF5 suppresses those genes that are highly expressed in spermatogonia to prevent their expression in spermatocytes.

In the revised manuscript, we conducted the similar analysis using robust dataset of HSF5-bound genes identified by CUT&Tag (new Figure 7d). This analysis provided a similar result to the former Fig. 61 in the previous manuscript. Among HSF5-bound genes identified by CUT&Tag, those genes assigned to class 2 were highly expressed in spermatogonial population (Clusters 11, 8, 4, 7, 5, and 6 populations in scRNA-seq), and those genes assigned to class 3 were highly expressed in earlier time points of meiotic prophase (Clusters 1, 0, and 9). Thus, HSF5 suppresses those genes that are highly expressed in spermatogonia and early meiotic prophase to prevent their expression at mid-pachytene stage onward.

Now we replaced the former Fig.61 (previous manuscript) with the new Fig.7d in the

revised manuscript.

What specific changes in the mRNA abundance of HSF5-bound genes occurs in pachynema? This seems like a critical missing component here – what is the outcome in terms of target genes in the absence of HSF5?

We added a new data on mRNA abundance of HSF5 target genes in the new Fig.7e.

We reanalyzed the mRNA abundance of HSF5-target genes (Class 1, 2, 3 shown in Fig. 7d) in pachynema using scRNA-seq data set, and compare overall mRNA abundance of HSF5-target genes between WT and *Hsf5*-KO.

This analysis revealed that ;

Class1 HSF5-target genes (related to male gamete generation, cilium organization, and intraflagellar transport), that were upregulated upon *Hsf5* expression, were downregulated in *Hsf5*-KO, suggesting that the Class1 genes were activated by HSF5.

Class 2 genes, that were highly expressed in spermatogonial populations, were upregulated in *Hsf5*-KO, suggesting that Class 2 genes were repressed by HSF5.

Class 3 HSF5-target genes (DNA metabolic process, mRNA metabolic process, and protein localization to organelle), that were downregulated upon *Hsf5* expression, were upregulated in *Hsf5*-KO, suggesting that Class 3 genes were repressed by HSF5.

What are the changes in chromatin during pachynema in *Hsf5* KO spermatocytes?

We examined the changes in chromatin during pachynema in *Hsf5* KO spermatocytes by ATAC-seq. This analysis indicated that overall chromatin accessibility is slightly increased at TSSs in HSF5-target genes in *Hsf5* KO. The same trend was observed in class 1,2,3 of the HSF5-target genes.

Thus, although we may discuss subtle changes in chromatin in *Hsf5* KO, we assume that assessing open chromatin in *Hsf5* KO versus control spermatocytes would add little new information to our paper because of limited interpretation the data. Therefore, this ATAC-seq data is solely to show the reviewers.

430-445 While quite interesting, these results are rather preliminary and are perhaps not the strongest to finish the results section with... what do they mean in terms of chromatin organization in pachynema? Can the authors show concomitant SWI/SNF chromatin-expected changes in pachynema?

We tried to validate the possible interaction of HSF5-SWI/SNF chromatin-remodeling factors by HSF5 IP-WB analysis, and found that SMARCA5, a SWI/SNF subunit, was under detection level by WB. One interpretation is that HSF5 interacts with SWI/SNF subunits weakly or transiently, consistent with lower detection levels of SWI/SNF subunits by MS spec analyses in the HSF5 immunoprecipitates. We discussed the possibility that the interaction of HSF5 with SWI/SNF could be regulatory rather than stoichiometric (Line 500-505).

Discussion

450-462 It is typical for this first paragraph to be a summation of the results of this study and their importance

We added a summation of the results in this study in the first paragraph in discussion part.

The authors should thoroughly highlight the advancements in this study in comparison to those from Bartuc et al., published in April 2023. As written, the authors minimize this previous study to the point where it almost seems like those authors didn't already present ~half of what is shown here

As described above, now we added discussion on differences between the two studies and advancements in this study in comparison to those from Bartuc et al (Line 523-559).

Reviewer #3 (Remarks to the Author):

This manuscript entitled “Atypical heat shock transcription factor HSF5 is critical for male meiotic prophase under non-stress conditions” by Yoshimura and colleagues investigate the mechanism underlying how HSF5 depletion in germ cells causes defects in spermatogenesis. The authors found the expression of this testis-specific heat shock protein HSF5 increased abruptly from the mid-pachytene spermatocytes, which coordinates its essential function in driving developmental progression beyond the mid-pachytene stage during spermatogenesis. The authors employed a range of methodologies including SMART RNA-seq, single-cell RNA-seq (scRNA-seq) and ChIP-seq to investigate the potential impact of HSF5 depletion on gene expression, and whether the genome-wide binding of HSF5 correlates with differentially expressed genes upon HSF5 knockout.

Many germ cell-specific proteins including a subset of transcription factors have been published about their requirement for meiotic progression beyond the pachytene stage, other HSF members have been reported to play a role in spermatogenesis under stress or non-stress conditions, and HSF5 has been shown to be essential for meiosis and male infertility in zebrafish (Saju et al. 2018) and mice (Barutc et al. 2023). Nevertheless, this study provides more mechanistic insights into the action of HSF5, and these interesting findings have the potential to constitute a valuable addition to the current knowledge.

This manuscript is overall well designed and well written. While many results have been convincing, the following concerns need to be addressed before further consideration of acceptance:

Major concerns

1. Lines 188-191: Since HSF5 protein is absent in KO testes, it needs to be explained why immunofluorescent signals are present in some KO seminiferous tubules and even not diminished compared to those in WT (Fig. 2C). If the antibody-raised signals are somehow not completely specific to HSF5 at all stages, the immunostaining results reflecting the stage-dependent cellular distribution of HSF5 should be reevaluated (Fig. 1F).

In the previous Fig. 2C, immunostaining was performed by HSF5-C antibody. We appreciated that our home-made HSF5-C antibody shows non-specific backgrounds along basement in seminiferous tubules and on chromosome axes. Now, we further

verified HSF5 immunolocalization patterns by other two different anti-HSF5 antibodies HSF5-N1 and HSF5-N2 (new Fig. S2a). Considering the consensus that were commonly observed by three different anti-HSF5 antibodies in WT and non-specific backgrounds observed in *Hsf5* KO, those IF experiments by HSF5-N1 and HSF5-N2 antibodies showed essentially same stage-dependent nuclear distribution of HSF5 as shown in the new Fig. 3c.

Furthermore, we verified HSF5 immunolocalization patterns by HA immunostaining using HSF5-3xFLAG-HA knock-in mice, which shows the same stage-dependent cellular distribution of HSF5 as that revealed by HSF5-N and HSF5-C antibodies (new Figure 2d). Therefore, the stage-dependent cellular distribution of HSF5 shown in the previous Fig. 1F meets the consensus as revealed by HSF5-C, HSF5-N1, HSF5-N2 antibodies and HA immunostaining in HSF5-3xFLAG-HA knock-in mice.

2. Lines 243-246: The spermatocytes in the selected images look like to be at late zygotene stage (Fig. 3E) - why not showing pachytene spermatocytes with an aberrant number of DMC1 foci in KO? The statistic result is inconsistent with about 50 DMC1 foci observed for each pachytene cell in WT (Luo et al., Nat Commun 2013).

Now, we replaced the images with the new ones showing pachytene spermatocytes in new Fig.4e.

3. Lines 355-364, Fig. 5: It remains unknown about the expression of autosomal and sex chromosome-linked genes in Cluster 10 population of *Hsf5* KO, since this subtype was largely absent in *Hsf5* KO testis. Although Cluster 2 population exhibited downregulation of sex chromosome-linked genes, the defects in meiotic silencing of the sex chromosomes might exist in *Hsf5* KO testis. Have the authors checked the distribution of MSC1 factors in H1T-positive mid-late pachytene spermatocytes using spread analysis?

Now we newly showed that Pol II was excluded from XY body in *Hsf5* KO pachytene spermatocytes (Fig. 4c). Further, we did not see upregulation of sex chromosomal genes in our SMART-seq data (Fig. S4) nor scRNA-seq data in *Hsf5* KO pachytene spermatocytes (Fig. 6j). Thus, our interpretation is that *Hsf5* KO pachytene spermatocytes formed the XY body and establish MSC1. We added our interpretation in the text (Line 244-251) and in the discussion (Line 523-559).

4. One intriguing finding of this manuscript is that scRNA-seq analyses reveal the expression levels of autosomal genes are elevated in *Hsf5* KO Cluster 2 subpopulation. Have the authors compared the average expression level of autosomal and sex chromosome-linked genes in *Hsf5* KO spermatocytes with wild-type cells from SMART RNA-seq datasets.

We have compared the average expression level of autosomal and sex chromosome-linked genes in *Hsf5* KO spermatocytes with wild-type cells from SMART RNA-seq datasets. However, we reasoned that there was a technical limitation in bulk RNA-seq

to compare average expression levels between WT and *Hsf5* KO, since subtle differences in developmental cellular populations between WT and *Hsf5* KO spermatocytes may have potentially contributed to bulk transcriptomic differences, as we stated (Line322-324). Indeed, scRNA-seq revealed that Cluster 2 and 10 populations of *Hsf5* KO spermatocytes were eliminated (Fig6c). Thus, it would add little information to show autosomal and sex chromosome-linked upregulated genes in *Hsf5* KO from SMART RNA-seq datasets.

5. Lines 364-367: As for the claim “...KO spermatocytes exhibited gene expression patterns that were comparable to those in WT...”, how about the gene expression differences seen as well from other Clusters (see those statistically meaningful based on the statistic labels), including sex chromosomal genes in Clusters 0 and 1 (Fig. 5I) and autosomal genes in Clusters 3 and 4 (Fig. 5J)? It’s stunning that the spermatocyte loss in Clusters 2 and 10 was due to gene expression differences at the same stage (Cluster 2) when HSF5 protein undergoes an expression burst rather than being due to those at earlier stages (prior to Cluster 2). Suggest to analyze deeply the gene expression profiles of earlier Clusters, especially Cluster 3, when cell populations were not yet sharply diminished.

We assessed the gene expression difference at the Cluster 3. Now we rephrased the relevant sentence as following : ”overall expression level of autosomal genes was downregulated in Cluster 3 subpopulation and then abruptly upregulated in Cluster 2 subpopulation of *Hsf5* KO” (Line 400-402).

“Thus, *Hsf5* KO spermatocytes exhibited gene expression changes at the mid-pachytene substage (Cluster 3) when HSF5 protein should undergo an expression burst, and consequently specific subpopulations of mid-pachytene” (Line 403-407).

6. It can be interesting to cross-analyze the SMART RNA-seq data and scRNA-seq data, as it’s reasonable to check all the 958 downregulated genes in *Hsf5* KO (SMART RNA-seq) through several related Clusters (scRNA-seq). How many genes are mutually inclusive between two datasets?

Now we cross-analyzed all the 958 downregulated genes identified in SMART RNA-seq

data (Fig. S4d, S4e) through Clusters in scRNA-seq data. This analysis revealed that 7% (Cluster 3), 69% (Cluster 2) and 92% (Cluster 10) of the top100 marker genes that are identified by scRNA seq were overlapped with the downregulated genes (SMART-seq) in *Hsf5* KO (see the Venn diagrams).

Similarly, averaged expression of all the 958 downregulated genes identified in SMART RNA-seq coincides with Cluster 2, 3, 9 and 10 (see the upper right UMAP).

This analysis revealed that 64% of the downregulated-genes in *Hsf5* KO, that are identified in Cluster 2, 3 and 9 by scRNA seq, were well overlapped with the downregulated genes in *Hsf5* KO by SMART seq.

The overlapping by these two different transcriptome analyses suggested that HSF5 positively regulated expression of those overlapping genes in the subpopulation of Cluster 2, 3 and 9.

Now we show the new data (new Fig. 6f) and stated this interpretation (Line 372-377).

7. Lines 372-377, Fig. 6A and B: Were total testis samples collected for ChIP-seq? HSF5 binding targets seems quite limited in testis compared with other typical transcription factors, what parameters are used for HSF5 genome-wide peak calling? These information should be added in the text or figure legend.

We collected whole testes for ChIP-seq as shown in the method. We used MACS with option (-g mm -p 0.00001) for HSF5 genome-wide peak calling. Now we added these informations in the figure legend (New Fig. S7a).

As the reviewer pointed out, HSF5 binding targets identified by our ChIP-seq were limited. Although we do not exact reason for this, we further validated HSF5 binding targets by another way of HSF5 CUT&Tag using isolated spermatocytes. Our CUT&Tag

analysis identified more HSF5 binding targets (2087 genes) in isolated spermatocytes. Further, we verified that our CUT&Tag data was robust enough to mostly encompass HSF5 binding targets identified by ChIP-seq (Fig S7d).

Now we added a new data of HSF5 binding targets identified by CUT&Tag in the new Figure 7.

Accordingly, we moved the ChIP-seq in the previous Figure 6 to the new Figure S7.

8. Lines 406-414: How could Hsf5 KO spermatocytes be rapidly eliminated at the pachytene stage, undergoing cell apoptosis or arrest to affect cell populations? The remaining cells of any same subtype surviving such elimination may be sufficient for transcriptomic analyses.

As we shown in Fig. 2e, HSF5 expression appeared just soon after H1t expression. In addition, as in Fig. 4a, H1t expressing cells were dramatically reduced in *Hsf5*-KO. Based on these results and Fig. 4g, *Hsf5*-KO cells were rapidly eliminated undergoing apoptosis.

Nevertheless, now we performed transcriptome analysis for a minor population of *Hsf5*-KO pachytene subpopulations that are before undergoing apoptosis. We reanalyzed the transcriptome of HSF5-target genes in the pachytene subpopulations (clusters 9, 3 and 2). This allowed us to compare overall expression levels of HSF5-target genes between WT and *Hsf5*-KO in the pachytene subpopulations where *Hsf5*-KO cells were still surviving before apoptotic elimination.

Our response to this query is also related the next question 9. Please see our response to the next question 9 for details.

9. Lines 416-428: Are any individual HSF5-bound genes (Classes C1 to C7), whether they are expressed at relatively high or low levels in any certain-stage germ cell population (Clusters 0 to 11) (Fig. 6K), dysregulated in Hsf5 KO at a given stage?

As mentioned above, we added a new data on gene expression levels of HSF5 target genes in the new Fig.7e.

First, we reanalyzed the alteration of mRNA levels of HSF5-bound genes that were identified by CUT&Tag using the scRNA-seq data from spermatogenic cells. Hierarchical clustering revealed the expression patterns of HSF5-bound genes across developmental direction were separated into 3 classes (Fig. 7d, Supplementary Data 6).

Further, we reanalyzed the expression patterns of HSF5-target genes (Class 1, 2, 3) using scRNA-seq data set, and compare overall expression levels of HSF5-target genes between WT and *Hsf5*-KO.

This analysis revealed that ;

Class1 HSF5-target genes (related to male gamete

generation, cilium organization, and intraflagellar transport), that were upregulated upon *Hsf5* expression, were downregulated in *Hsf5*-KO, suggesting that the Class1 genes were activated by HSF5.

Class 2 genes, that were highly expressed in spermatogonial populations, were upregulated in *Hsf5*-KO, suggesting that Class 2 genes were repressed by HSF5.

Class 3 HSF5-target genes (DNA metabolic process, mRNA metabolic process, and protein localization to organelle), that were downregulated upon *Hsf5* expression, were upregulated in *Hsf5*-KO, suggesting that Class 3 genes were repressed by HSF5.

10. Lines 430-436: The immunoprecipitation and mass spectrometry analysis with three different HSF5 antibodies identified that SMARCA5 and SMARCC2 proteins interacted with HSF5. This result needs to be confirmed by Co-IP using in vivo or in vitro system. We tried to validate the possible interaction of HSF5-SWI/SNF chromatin-remodeling factors by HSF5 IP-WB analysis, and found that SMARCA5, a SWI/SNF subunit, was under detection level by WB. One interpretation is that HSF5 interacts SWI/SNF subunits weakly and/or transiently, consistent with lower detection levels of SWI/SNF subunits by MS spec analyses in the HSF5 immunoprecipitates. This is also the case for HSF5-KCTD19 interaction. Although these data were not strong enough to account for the potential role of HSF5 in the gene activation and repression, we thought the HSF5 IP-MS data are still informative for the field. Thus, we discussed the possibility that those interactions of HSF5 with SWI/SNF and KCTD19 could be regulatory rather than stoichiometric in the discussion (Line 500-505, Line 512-518).

Minor concerns

1. Line 59: The citation (Kojima et al. 2019) is missing in the reference list.

We corrected this.

2. Lines 119-120: As introduced by the authors, the biological function of HSF5 has already been reported elsewhere.

We added the following sentence “the biological function of HSF5 was yet to be fully elucidated in mice except that HSF5 is required for establishment of male meiotic sex chromosome inactivation (Barutic et al. 2023)”.

3. Fig. 1F: Some images look weird, for instance, the inner cells stained as red (for SYCP3) should be non-specific signals at stages IV-V and VI.

The gunia pig anti-SYCP3 antibody that we used in Fig. 1F shows non-specific signal to sperm tail in the condition of the immunostaining. Thus, we indicated the non-specific immunostaining by * in the panel of stages IV-V and VI in Fig. 1F, and noted * indicates

a non-specific cross reactivity to sperm tail in the legend.

4. Fig. 3F: Are the labels reversed between SYCP3 and MLH1? Red foci should be MLH1 and not clearly seen on the images.

Labels for SYCP3 and MLH1 were corrected. Immunostained MLH1 signals were overall weak. For clarity, the contrast in red channel of weak MLH1 signals was enhanced.

5. Lines 322-330: What were the germ cell subtypes designated for Clusters 4, 5, and 6? Cluster 4 is followed by Cluster 7, and Cluster 7 is followed by Clusters 5 and 6. Since Cluster 7 represents the population of differentiating spermatogonia, it is assumed that Cluster 4 represented the population of spermatogonia A. Clusters 5 and 6 represented the population of spermatogonia B.

We added those sentences in the text (Line 354-356).

6. Lines 339-341: In Cluster 2, spermatocytes seem NOT totally absent in KO (Fig. 5C). We rephrased as following “the subtype of spermatocytes that represented Cluster 2 was largely less frequently observed and that of Cluster 10 was absent in *Hsf5* KO spermatocytes” (Line 369-371).

7. Lines 355-358: In Cluster 10, there seems very few spermatocytes that can be analyzed in KO (Fig. 5C), thus “abrupt downregulation” may be not properly claimed. Then, is it logical to suggest that “those Cluster 2 represents...”?

We rephrased the sentence as following “overall expression level of the sex chromosome genes was gradually down-regulated in the progression to Cluster 10 through Cluster 2 (Fig. 6j), suggesting that Clusters 2 and 10 represented the transition when meiotic sex chromosome inactivation (MSCI) is going to be established.

8. Lines 367-369: Suggest to rewrite this conclusion towards the angle of gene expression regulation by HSF5.

We revised the conclusion as following “HSF5 play a role in regulating gene expression programs for the developmental progression beyond the mid-pachytene substage during spermatogenesis”.

9. Discussion: Suggest to reduce redundant comments, for example, lines 496-498.

We removed redundant sentences.

REVIEWERS' COMMENTS

Reviewer #1 (Remarks to the Author):

Authors should include the ATAC-seq data that they present to the reviewers into the manuscript. Otherwise, the authors have addressed my comments satisfactorily.

Reviewer #3 (Remarks to the Author):

The authors have addressed all the comments with additional analyses and experiments, the manuscript is now fit for publication in Nature Communications.

NCOMMS-23-42503 REVIEWER COMMENTS

Reviewer #1 (Remarks to the Author):

Authors should include the ATAC-seq data that they present to the reviewers into the manuscript. Otherwise, the authors have addressed my comments satisfactorily.

Now, we presented the ATAC-seq data in the new supplementary Figure 7, and accordingly revised the main text (Line 419-423) and method section (Line 1088-1128).

Reviewer #3 (Remarks to the Author):

The authors have addressed all the comments with additional analyses and experiments, the manuscript is now fit for publication in Nature Communications.